# The Applications of Polymers in Solar Cells: A Review

**DOI:** 10.3390/polym11010143

**Published:** 2019-01-15

**Authors:** Wenjing Hou, Yaoming Xiao, Gaoyi Han, Jeng-Yu Lin

**Affiliations:** 1Institute of Molecular Science, Key Laboratory of Chemical Biology and Molecular Engineering of Education Ministry, Key Laboratory of Materials for Energy Conversion and Storage of Shanxi Province, Innovation Center of Chemistry and Molecular Science, Shanxi University, Taiyuan 030006, China; wjhou2016@sohu.com; 2Department of Chemical Engineering, Tatung University, Taipei 104, Taiwan

**Keywords:** applications, polymers, solar cells

## Abstract

The emerging dye-sensitized solar cells, perovskite solar cells, and organic solar cells have been regarded as promising photovoltaic technologies. The device structures and components of these solar cells are imperative to the device’s efficiency and stability. Polymers can be used to adjust the device components and structures of these solar cells purposefully, due to their diversified properties. In dye-sensitized solar cells, polymers can be used as flexible substrates, pore- and film-forming agents of photoanode films, platinum-free counter electrodes, and the frameworks of quasi-solid-state electrolytes. In perovskite solar cells, polymers can be used as the additives to adjust the nucleation and crystallization processes in perovskite films. The polymers can also be used as hole transfer materials, electron transfer materials, and interface layer to enhance the carrier separation efficiency and reduce the recombination. In organic solar cells, polymers are often used as donor layers, buffer layers, and other polymer-based micro/nanostructures in binary or ternary devices to influence device performances. The current achievements about the applications of polymers in solar cells are reviewed and analyzed. In addition, the benefits of polymers for solar cells, the challenges for practical application, and possible solutions are also assessed.

## 1. Introduction

The rapidly growing population and world economy have led to the continuously growing energy demand [1]. The enormous consumption of traditional fossil energy has resulted in the serious resource exhaustion and environmental pollution. One of the most imperative solutions is to look for alternative renewable energies. The solar energy stands out owing to solar is environmentally friendly and free from regional restrictions [1,2]. The solar cell is one of the most effective utilization approaches for solar energy. At present, the research and development of solar cells mainly focus on: (1) Mature silicon-based solar cells. Although the laboratory efficiency can reach more than 25% [3], the development of traditional silicon-based solar cells was limited by the sophisticated manufacturing process, high energy consumption and high cost [4]. (2) Thin-film solar cells. The typical thin-film solar cells include the gallium arsenide (GaAs) [5], copper indium gallium selenide (CIGS) [6], and cadmium telluride (CdTe) [7], etc. Although these thin-film solar cells have high efficiency and stability, the abundance of gallium and indium in the crust is low and cadmium is toxic. (3) Emerging solar cells. The emerging solar cells are represented by organic photovoltaics (OPV) [8], dye-sensitized solar cell (DSSC) [9] and perovskite solar cell (PSC) [10]. This kind of solar cells have the advantages of lightweight and low cost. However, there are some problems that limit their large-scale application. For example, organic solar cells still need to be improved in conversion efficiency, spectrum response range and device stability [11]. Dye-sensitized solar cells based on liquid electrolyte are faced with the problems of easy electrolyte leakage and solvent evaporation, which undermine the long-term stability of the device [12]. The all-solid perovskite solar cell, which develops on the basis of DSSC but overcomes the shortcomings of DSSC, has attracted extensive attention from scientists as soon as they were reported. Even though the photoelectric conversion efficiency (PCE) of PSC has ranged from initial 3.8% to the latest 23.3% rapidly **(**Figure 1**)**, the device stability is still a problem that limited the applications [13,14,15]. Generally, each component of the device influences its performances, by performing their respective tasks. Hence, optimizing the device components is of great significance.

In the past few years, the polymers have been studied widely due to their versatile and adjustable chemical and physical properties [16]. The three-dimensional network structures of polymers decide that they can be employed as the template to fabricate mesoporous materials or be used as a polymeric matrix in solid electrolyte [17,18,19]; the high catalytic activity for I_3_^−^ reduction make it potential as counter electrode materials. The diversified functional groups make it possible for the polymers to regulate the perovskite morphologies from bulk and interface aspects; the high carrier mobilities enable polymers acting as electron and hole transfer materials; the functional groups in polymers decide that they can be used as the interface layers to passivate defects, adjust the work function of the metal electrode, and improve the device performances [20,21]; the diversified structures and functional modification also equip polymers with various optical adsorption properties and variable electron mobility, being used as the photo-active layer or buffer layer in OPV [22,23]; The processability of polymer also make it possible to fabricate polymer-based micro/nanostructure devices. The partial polymers with good conductivity, named as conductive polymers, were widely used in many fields [24,25,26].

In this review, the applications of polymers in solar cells are mainly concentrated on DSSC, PSC, and OPV. 

## 2. Polymers in DSSC

Inspired by the photosynthetic process in plants, the O’Regan and Grätzel proposed the first prototype of DSSC in 1991 [27]. The DSSC has achieved stunning progress in recent years due to its low cost, excellent efficiency, and handy and eco-friendly fabrication process [28]. In DSSC, the electrolyte was encapsulated between the anode and the counter electrode to form a sandwich structure. The principle of DSSC was shown in Figure 2 [29,30,31,32,33]. Generally, the electrons at excited state inject the conduction band (CB) of semiconductor, and then flow to the counter electrode through the external circuit. The oxidized electrolyte obtains the electrons from the counter electrode to produce the reduced electrolyte, realizing the electrolyte regeneration; The electron recombinations occur at these three parts and their interfaces. In this sense, the three parts of the DSSC play the imperative and indispensable roles in deciding the device performances. As we all know, the choosing and fabricating of constituent materials are crucial. Due to the various properties, the polymers can be applied into fabricating the constituent materials, such as: The polymers can be used to fabricate flexible substrates, to constitute mesoporous structure in anode, to prepare polymer gel electrolyte, and to catalyze electrolyte reduction as counter electrodes. The applications of polymers in DSSC are illustrated in the following sections in detail.

### 2.1. Polymers as the Substrates of DSSC

Based on the principles of DSSC, the conductive substrate is responsible for supporting cells, transmitting light, collecting and transmitting electrons. The common conductive substrates include indium-doped tin oxide (ITO) glass, fluorine-doped tin oxide (FTO) glass, ITO/polyethylene terephthalate (PET), and ITO/polyethylene naphthalate (PEN). The ITO/PEN and ITO/PET are the most common conductive polymer substrates in DSSCs due to their low cost, high transparency, lightweight, flexibility, and low sheet resistance about 10–15 Ω·sq^−1^ [34,35,36,37]. These merits of ITO/PEN and ITO/PET make it possible to fabricate the roll-to-roll solar cells and other flexible and wearable electronic devices [37]. In DSSCs, the substrates loaded with TiO_2_ films often need to be sintered at 450–500 °C to guarantee the high crystallinity and purity of TiO_2_. And the high temperature sintering also ensures the tight contact between the TiO_2_ films and the conducting substrates [35,38]. Besides, the temperature above 400 °C also is applied to the pyrolysis of platinum (Pt) precursor in the fabrication of Pt counter electrode [38]. Nevertheless, the ITO/PEN and ITO/PET substrates are always sensitive to the high temperature [38]. Once the processing temperature exceeds 150 °C, the ITO/PET and ITO/PEN substrates begin to deform, and even melt at 235 °C [38,39]. The restricted processing temperature below 150 °C goes against the improvement of contact performances between conductive polymer substrates and TiO_2_ anode. These drawbacks inspire the developments of other low temperature deposition techniques. 

### 2.2. Polymers in Mesoporous TiO_2_ Photoanode of DSSC

The device performances of DSSC also depend on the photoanode largely. The photoanode in DSSC surves as the support for dye-absorbing, where the incident light is harvested and the electron transport takes place [40]. One of the most common photoanode materials is TiO_2_ due to its photochemical stability, nontoxicity, and adequate resources for synthesis [41,42,43]. 

In order to improve the dye sensitization effect of TiO_2_ nanoparticles photoanode, TiO_2_ films are required to have a large specific surface area, which can be realized by fabricating mesoporous TiO_2_ film. The more pore structure is, the larger specific surface area is [43,44]. Benkstein and co-workers have reported that the porosity in photoanode film can be realized by applying polymers as pore-forming regent [45]. In the subsequent annealing process, the polymers in photoanode films can be sintered, leaving the pores in their original positions [31,32,45]; and the content of polymers in the photoanode film can be used to control the porousness, which is ideally about 50−60% [31,46]. Excessive polymers create too many pores, which reduce the interconnects between particles and charge collection efficiency [46]. Hou et al. fabricated a series of mesoporous TiO_2_ anodes by annealing the blade-coated TiO_2_ films, which contain different content polyvinylpyrrolidone (PVP) [47]. The device based on the TiO_2_ anode with 20 wt % PVP gives the highest conversion efficiency of 8.39%, approximately 56.63% higher than that of without PVP (5.36%). The enhanced efficiency can be ascribed to that moderately increased porosity can expose more specific surface areas, increasing the dye loading and enhancing the electron transfer and electrolyte diffusion efficiency further. The excessive PVP leads to small specific surface areas owing to excessive PVP reduced the connect between TiO_2_ nanoparticles and increased the pore volume. Yun et al. synthesized two kinds of mesoporous TiO_2_ photoanodes (MP-TiO_2_ (A) and MP-TiO_2_ (B)) by using tri-block copolymer (polyethylene oxide-polypropylene oxide-polyethylene oxide, P-123) and inorganic ZnO/Zn(OH)_2_ templates, respectively [48]. The device based on the MP-TiO_2_ (A) achieves the efficiency of 6.71%, higher than that based on the MP-TiO_2_ (B) (3.05%) and pure P25 (5.62%). The best performances of MP-TiO_2_ (A) based device can be attributed to the largest surface area, the largest pore size, and the largest pore volume of MP-TiO_2_ (A) photoanode than other referential photoanodes [48], which can be benefited from the P-123 template. However, the electron mobility in TiO_2_ nanoparticle systems is slow due to possible energy loss in the electron trapping process, which increases recombination risks in mesoporous photoanode films [49]. 

In order to solve above problem, the one-dimensional TiO_2_ photoanodes (such as TiO_2_ nanofibers and nanotubes) have been widely used in DSSCs due to their special feature: Directional electron transport along the fibers or tubes, and the reduced scattering effect [50]. The electrospinning method is initially used to fabricate the one-dimensional materials, including the one-dimensional TiO_2_. Kokubo fabricated the multi-core cable-like TiO_2_ nanofibrous membrane by electrospinning the dimethylformamide solution containing 11.5 wt % of polymer vinyl acetate (PVAc) and 1 g titanium isopropoxide and then calcinating the composite of PVAc/titania nanofibres at 500 °C [51]. Then the TiO_2_ nanofibrous membrane was used as the photoanode of DSSC, obtaining the efficiency of 5.77%. Joshi fabricated the TiO_2_ nanofiber photoanode by electrospinning the solution of isopropanol/demethylformamide containing Titanium n-butoxide (TNBT) and polyvinylpyrrolidone (PVP) and then pyrolysising [52]. The DSSC based on the TiO_2_ nanofiber achieved the efficiency of 2.9%. The low efficiency can be deduced to the low dye uptake of 30.49×10^−6^ mol·g^−1^, which can be attributed to the small specific surface area of TiO_2_ nanofiber caused by large diameters [52]. In view of this problem, Joshi et al. prepared the composite TiO_2_ nanofibers/nanoparticles (TiO_2_ NPs/NFs) photoanode by electronspinning the PVP/TNBT nanofibers on the as-prepared TiO_2_ nanoparticles and calcinating them, which obtains a greatly enhanced conversion efficiency of 8.8% [52]. He et al. prepared the TiO_2_ photoanode with nanotubes structures by co-axial electrospinning the precursor solution of TNBT and PVP. The device based on the TiO_2_ nanotubes photoanode can realize the efficiency of 3.80% [53].

Therefore, we can conclude that the polymers can be used to regulate the electron transfer in photoanode by influencing the porousness and morphologies of photoanodes.

### 2.3. Polymers as the Counter Electrodes of DSSC

The counter electrode (CE) in DSSC is responsible for catalyzing the reduction reaction of I_3_^−^, deciding the device performances further [54,55]. The excellent catalytic activity for I_3_^−^ reduction equips the Pt serving as the most common counter electrode material [54]. However, the using of Pt CE hampered the further development and long-term stability of Pt-based DSSC due to its scarcity and possible corrosion in I_3_^−^/I^−^ electrolyte system [56]. Therefore, the overwhelming majority of research focuses on looking for alternative materials of Pt CE, such as carbon materials [57,58], metal alloys [59,60], transitional-metal compounds [61,62], conductive polymers [63,64,65,66,67,68] and corresponding composite [69,70,71,72]. Compared with other alternatives, the conductive polymers possess the properties of low cost, high catalytic activity, translucence, facile synthesis, and abundance, which decide the conductive polymers can act as one of the most ideal alternatives of Pt CE [63,64,65,66,67,68]. The polypyrrole, polyaniline, and poly(3,4-ethylenedioxythiophene) are commonly used as conductive polymers CEs materials.

#### 2.3.1. Polypyrrole as the Counter Electrodes of DSSC

The easy synthesis process, high yield, favorable catalytic activity, and environmental stability make the polypyrrole (PPy) be a potential alternative of Pt CE [73,74,75,76]. Table 1 summarizes the photovoltaic performance parameters of DSSCs based on the PPy CEs. The PPy is often formed by polymerizing the pyrrole monomer, accompanying the color changing from yellow to blue or black. The polymerizations can be divided into chemical polymerization and electrochemical polymerization. For the chemical polymerization of PPy, the various oxidants can be used. Peng et al. synthesized the free-standing paper-like polypyrrole nanotube membrane CE by using the Iron(III) chloride hexahydrate (FeCl_3_·6H_2_O) to oxidize the pyrrole monomer (Figure 3a,b). The device based on the PPy nanotube membrane counter electrode obtains the efficiency of 5.27%, being comparable with that of Pt CE (6.25%) [76]. Wu and co-workers fabricated the PPy CE by coating the iodine oxidation polymerized PPy nanoparticles on FTO. The device assembled with the PPy CE obtained a PCE of 7.66%, higher than that of Pt CE (6.90%) [77]. Bu et al. fabricated bifacial DSSC assembled with the transparent and stable PPy CE and giving the efficiency of 5.74%. In this case, the FTO was immersed into the mixed solution of the ammonium persulfate (APS) and pyrrole monomer at 0 °C for 3 h, where the APS was used as the oxidant [78]. Besides, Wang fabricated PPy CE with hierarchical nanostructure by using the vanadiumpentoxide (V_2_O_5_) nanofiber and FeCl_3_ to oxidize the pyrrole monomer step by step. The best performing device realizes the efficiency of 6.78%, higher than that of PPy nanoparticles (5.41%), being 92.5% of Pt CE (7.33%) [79]. Besides, Xu et al. prepared the PPy-coated cotton fabrics CE by electrochemical polymerization, and the corresponding device obtains the efficiency of 3.83% [80]. Even copious amounts of research focus on the PPy CE, the device efficiencies based on the PPy CEs are disadvantaged. The main issues are concentrated on the high charge transfer resistance (*R_CT_*) and low conductivity of PPy [81].

#### 2.3.2. Polyaniline as the Counter Electrodes of DSSC

Except PPy, the polyaniline (PANI) is also the most studied conductive polymers due to its relatively low cost, multiple oxidation states with different colors, and acid/base doping responsiveness. Table 2 lists the photovoltaic performance parameters of DSSCs based on the PANI CEs. Li and Wu et al. prepared the microporous PANI CE with the average size diameter of 100 nm by perchloride acid doped APS oxidative polymerization. The device with PANI CE realized the efficiency of 7.15%, higher than that of Pt CE (6.90%) [82]. Compared with the random transport in PANI nanoparticles CE, the oriented morphologies are conducive to accelerate the electron transfer efficiency and enhance the device performances further. As Figure 3c,d shown, Hou et al. synthesized a polyaniline nanoribbon (PANI NR) CE with serrated, flexible and ultrathin nanostructures by using the electrospun V_2_O_5_ as template and oxidant to realize the in situ polymerization of aniline and etching the V_2_O_5_ template by acid [83]. The lots of active sites, good contact performance with substrate, and the oriented electron transfer along the nanoribbons equip the PANI NR based DSSC a PCE of 7.23%, which can be comparable to that of the Pt-based DSSC (7.42%). Wang et al. in-situ fabricated the oriented PANI nanowire counter electrode by immersing the FTO glass into the aniline/APS solution with the fixed mole ratio of 1.5 at 0–5 °C for 24 h. The obtained PANI nanowire CE exhibits higher cataltic activity and device performances (8.24%) than that of random PANI nanofibers (5.97%) and Pt CE (6.78%) [84]. The enhanced performance can be deduced to the effectively exposed PANI nanowires and the rapid electron transfer along the oriented nanowire. Besides, Xiao et al. fabricated the PANI CE with nanofiber structure by using the pulse potentiostatic electropolymerization method, giving the efficiency of 5.19%, up to 90% of Pt CE [85]. 

The previous research has indicated that the doping ions will influence the morphologies of electrode materials and corresponding electrochemical properties [86]. Li et al. compared the performances of different PANI films doped by various counter ions SO_4_^2−^, ClO_4_^−^, BF_4_^−^, Cl^−^, and p-toluenesulfonate (TsO^−^), etc. And the results revealed that the DSSC based on the PANI-SO_4_^2−^ CE shows the efficiency of 5.6%, being comparable to that of Pt CE (6.0%) [87]. 

Even though PANI possesses the superior catalytic activities, high conductivity, simple fabrication process, the applications of PANI CE are still limited due to its carcinogenic and self-oxidation properties [88]. 

#### 2.3.3. Poly(3,4-ethylenedioxythiophene) as the Counter Electrodes of DSSC

As another important conductive polymer, the poly(3,4-ethylenedioxythiophene) (PEDOT) presents the highest catalytic activity for I_3_^−^ reduction and the highest conductivity (300–500 S·cm^−1^) than those of PANI (0.1–5 S·cm^−1^) and PPy (10–50 S·cm^−1^) [89]. The existing conductivity has reached more than 4600 S·cm^−1^ by doping poly(styrene sulfonate) (PSS) [90]. Except for the high conductivity and catalytic activity, the excellent thermal and chemical stability, and electrochemical reversibility have made the PEDOT be an ideal material for DSSC CEs [90,91]. Table 3 enumerates the photovoltaic performance parameters of DSSCs based on the PEDOT CEs. Li et al. fabricated honeycomb-like PEDOT CE by a facile cyclic voltammetry electrodeposition and sacrificial template methods. The resultant device based on the honeycomb-like PEDOT CE exhibits the front efficiency of 9.12% and rear efficiency of 5.75%, outperforming than that of flat PEDOT (8.05% and 3.78%, respectively), which lights up the roads of bifacial and tandem devices [92]. Trevisan et al. fabricated the PEDOT nanotube arrays (Figure 3e,f) by combining the electrodeposition and template etching technique. The best performing device yielded the efficiency of 8.3% [93]. Xia et al. studied the influence of doping ions on the morphologies of PEDOT and corresponding device performances. The ClO_4_^−^ and PSS doped PEDOT CE presented higher efficiencies than those of TsO^−^ doped PEDOT [94].

#### 2.3.4. The Hybrids Based on Conductive Polymers as the Counter Electrodes of DSSC

Even the conductive polymers present the high catalytic activity for I_3_^−^ reduction, the conductivity of them are still limited due to their semiconductor nature [89]. Therefore, the hybrids based on the conductive polymers have been studied to improve the device performances (Figure 4 and Table 4). The hybrids make the utmost of the synergetic effects of corresponding components. The conductive polymers-based hybrids include the following three parts: (1) Conductive polymers/carbon materials [95,96,97,98,99,100,101]; (2) conductive polymers/transitional metal compounds [102,103]; and (3) conductive polymers/conductive polymers [104,105].

The carbon materials are used for the hybrids due to their excellent conductivity, such as single/multi-wall carbon nanotube (SWCNT/MWCNT), graphene, and carbon black nanoparticles, and carbon nanofibers, etc. [96,97,98,99,100,101]. The composites of conductive polymers/carbon materials combine the excellent conductivity of carbon materials with the high catalytic activity of conductive polymers, which accelerate the charge transfer in counter electrodes. He et al. fabricated the composite CE of PANI/graphene by electrodepositing the aniline-graphene complex. The device based on the PANI-8 wt ‰ graphene CE displayed the efficiency of 7.70%, which is superior to that of pure PANI CE (6.40%) [96]. Zhang et al. prepared the PANI-8 wt ‰ MWCNT CE with high transmittance by using the electrodeposition methods, achieving a front efficiency of 7.91%, rear efficiency of 1.72%, and the maximum efficiency of 9.24%, higher than that of pure PANI CE [97]. Compared to pure PPy CE, the DSSC based on the PPy-2 wt ‰ SWCNT CE also presented the enhanced short current density and efficiency [98]. Xiao’s group systematically prepared a series of composite CEs of conductive polymers and MWCNT (PANI/MWCNT, PEDOT/MWCNT, and PPy/MWCNT), which yielded the enhanced efficiencies compared to their corresponding component materials, respectively [99,100,101].

Xu and Cui et al. synthesized the TiN-PEDOT:PSS hybrid CE by simple physical mixing and scraping. The device with the TiN-PEDOT:PSS hybrid CE obtained the enhanced performances owing to that the TiN-PEDOT:PSS hybrid CE ensures the good contact between the TiN-PEDOT:PSS and FTO substrate and avoids the TiN from aggregating [102]. Di et al. fabricated the composite CEs of transitional metal phosphates (Ni_3_(PO)_4_, Co_3_(PO)_4_, and Ag_3_(PO)_4_) and PEDOT. The device assembled with the Ni_3_(PO)_4_-PEDOT CE realized the efficiency of 6.412%, higher than that of Co_3_(PO)_4_-PEDOT CE (6.109%), Ag_3_(PO)_4_-PEDOT CE (3.731%), and original PEDOT CE (5.443%) [103]. The composite PEDOT:PSS/PPy CE was synthesized and the device based on the PEDOT:PSS/PPy CE gave the efficiency of 7.60%, higher than that of PEDOT:PSS (6.31%) and PPy (5.23%), and 98.3% of Pt (7.73%) [105].

These hybrid CEs take the advantage of the high catalytic activity of conductive polymers, the high conductivity of carbon materials, the large specific surface area of transitional metal compounds, hence improving the device efficiency. 

### 2.4. Polymers as the Electrolyte of DSSC

As one of the most imperative components in DSSCs, the electrolyte undertakes the responsibility for transferring carrier, regenerating the dye and itself [106]. The electrolyte exerts dominated influences on the device performances by influencing the related performance parameters, such as the short current density (*J_SC_*), open-circuit voltage (*V_OC_*), and fill factor (*FF*) [29,107,108]. Specifically speaking, the transfer rate of redox couple in electrolyte will affect the *J_SC_* [29,107]. The charge transfer impedance at the interfaces of electrolyte/electrodes has non-ignorable impacts on the *FF* [29,107]. The *V_OC_* is a function associated with the femi energy level of the photoanode and the redox potential of the electrolyte [29,108]. Hence, the research about the electrolyte are necessary. The charge transfer in electrolyte is controlled by diffusion, which can be affected by the components, the concentration of redox couple, the viscosity of the solvent, and the distance between photoanode and counter electrode [29,109].

In 1991, O’Regan and Grätzel first employed the organic solvent containing the redox couple (such as I^−^/I_3_^−^) as the electrolyte in DSSCs and achieved an efficiency of 7.0–7.9% [27]. Since then, the liquid electrolytes have been utilized universally. The devices with high efficiency (13%) and champion efficiency (14.3%) are also based on the liquid electrolyte [110,111,112]. The corrosion of the CE materials, and the degradation and desorption of dye all have a great relationship with the liquid electrodes [113,114]. Meanwhile, the leakage of liquid electrolytes and the volatilization of the solvent all damage the device performances and stability [113,114]. The quasi-solid-state polymer electrolytes are promising to serve as the alternatives of liquid electrolyte, because they are conducive to improve the sealing and stability [29,114,115,116]. The quasi-solid-state electrolyte are often fabricated by solidifying the liquid electrolyte solution using the organic polymer gelatin [29,114]. Therefore, the organic solvent, polymer or oligomer, and inorganic salt are the indispensable components of quasi-solid-state electrolyte [29]. According to the types of interaction in electrolyte, the polymers-based quasi-solid-state electrolytes can be divided into the thermoplastic polymer electrolyte, thermosetting polymer electrolyte, and composite polymer electrolyte [29], which are widely used in the DSSCs (Table 5).

#### 2.4.1. Thermoplastic Polymers as the Electrolyte of DSSC

For the thermoplastic polymer electrolyte, the polymer or oligomer fabricates the polymer framework firstly. After mixing the liquid electrolyte with the polymer framework, the system gradually becomes into a viscous gel state under the stimulus of temperature, where the gelatin, adsorption, inflation and polymer assembling processes occur [29,117]. The solvent in liquid electrolyte diminishes the interactions between polymer chains by forming the van der waals and electrostatic interaction with polymers [29,117]. These weak interactions permit the moderate solvent keeping in the gel, presenting the excellent interfacial wetting and filling properties, and high ionic conductivity [29,117,118]. Meanwhile, limited by the polymer matrix, the thermoplastic polymer electrolyte shows the merits of solid electrolyte.

The common polymer gelatin in thermoplastic polymer electrolyte include the poly(acrylonitrile) (PAN), poly(ethylene oxide) (PEO or PEG), PVP, poly(vinylidene ester) (PVE), polystyrene (PS), poly(vinyl chloride) (PVC), poly(methyl methacrylate) (PMMA), and poly(vinylidene fluoride) (PVDF), etc. Cao et al. first fabricated the thermoplastic polymer electrolyte by introducing the I_2_/NaI-based liquid electrolyte into PAN polymer [119]. The device assembled with this electrolyte obtains the excellent performances and stability. Wu et al. employed the liquid electrolyte of ethylene asrbonate (EC)/prpopylene carbonate (PC)/N-methyl pyridine iodide and poly(acrylonitrile-*co*-stryrene) matrix to fabricate the gel state electrolyte and obtained the efficiency of 3.10% [120]. Wu et al. synthesized the PC/PEG/KI/I_2_ gel electrolyte and achieved the efficiency of (7.22%), being competed to that of liquid electrode (7.60%) [121]. The functional groups make PEO and PEG enhancing the ionic conductivity by forming the interaction with the alkali metal cations and leaving the free-moving iodide anions, thus reducing the recombination and enhancing the device performances [122]. Lee et al. injected the liquid electrolyte into the gap between photoanode and couner electrode coated with PS (Figure 5). The PS is dissolved and becomes into gel state once uptaking the liquid electrolyte. The device based on this gel electrolyte and the liquid electrolyte exhibited the similar efficiency of 7.59% and 7.54%, respectively [123].

Even the utilizations of gel-based electrolytes improve the device stability by avoiding the electrolyte leakage, it is still dilemma to control the ratios of polymers and liquid electrolyte [29,124]. Excessive polymers host will retard the ionic movement [29,124]. Excessive liquid electrolyte will lead to the sealing problem.

#### 2.4.2. Thermosetting Polymers as the Electrolyte of DSSC

For thermosetting polymer electrolyte, the states cannot change with temperature. Under the stimulus of light or heat, the organic molecules in the mixed solution in-situ cross-link to form the thermosetting polymer electrolyte through chemical covalent interaction, in which, the liquid electrolytes were wrapped with entangled polymer matrix [29,125]. This method can help the electrolytes in-situ fill into the gel electrolyte. Besides, swelling the cross-linked macromoleculor polymer in liquid electrolyte is another representative method to prepare the thermosetting polymer electrolyte [29,126]. For the DSSC device, the electrolyte filling in the electrodes is of great significance. Compared with the liquid electrolyte, it is difficult for the gel electrolyte to impregnate into the mesoporous TiO_2_ film [127]. Parvez et al. infiltrated mesoporous TiO_2_ film with the polymer monomer solution firstly [127]. After treating by the UV light illumination, poly(ethylene glycol) (PEG) and poly(ethyleneglycol) diacrylate (PEGDA) formed the crosslinked gel structure without damaging the device components. The efficiency and the long-term stability are all enhanced. Wang et al. prepared a novel necklace-like polymer gel electrolyte by heating the precursors of PPDD/I_2_/I^−^/NMBI/TBP and obtained the efficiency of 7.72%, where the PPDD is polypyridyl-pendant poly(amidoamine) dendritic derivatives, NMBI reprents the *N*-methylbenzimidazole, TBP refers to the tert-butylpyridine [128]. As depicted in Figure 6, Wu et al. synthesized the PEG/PAA/I_2_/I^−^ gel electrolyte by taking advantage of the amphipathy of PEG and the superabsorbent of PAA to absorb the liquid electrolyte. The device with the PEG/PAA/I_2_/I^−^ gel electrolyte achieved an efficiency of 6.10%, which is comparable to that based the liquid electrolyte (6.70%) [129].

Despite tremendous efforts have been devoted in gel electrolyte, the *J_SC_* of devices based on polymer gel electrolytes are still lower than that assembled with liquid electrolyte due to the limited conductivity [129].

#### 2.4.3. Composite Polymer as the Electrolyte of DSSC

Apart from the thermoplastic and thermosetting polymer electrolytes, the composite polymers are also one kind of the most common electrolytes. The copmosite polymer electrolytes are fabricated by adding the inorganic nanoparticles into the liquid electrolyte containing the polymers, where the inorganic nanoparticles serve as the gelatins to solidify the liquid electrolyte and to enlarge the amorphous phase in electrolyte [29,130]. In this type of electrolytes, the introducing of inorganic nanoparticles can enhance ionic conductivity and device stability. The fabrication of PEO/TiO_2_/LiI/I^−^ composite electrolyte creates the space for I^−^/I_3_^−^ migration due to the arrangement of TiO_2_ in the polymer network [131]. Yoon et al. used the spherelike and rodlike SiO_2_ as the gelatins of poly(ethylene glycol) dimethyl ether (PEGDME) oligomer (Figure 7) [132]. The device performances were improved due to the enhanced redox couple diffusion. Huo et al. introduced TiO_2_ nanoparticles into Poly(vinylidenefluoride-*co*-hexafluoropropylene) P(VDF-HFP) based gel electrolyte. After introducing TiO_2_ nanoparticles, the ionic diffusion coefficient of I_3_^−^ in P(VDF-HFP) based gel electrolyte increased six times and could be comparable to that in liquid electrolytes. Ultimately, the best performing device reached the efficiency of 7.18%, even higher than that of liquid electrolyte-based DSSCs. After heating at 60 °C for 1000 h, the best performing device maintains 90% of its initial efficiency [133].

### 2.5. Polymers in All-Weather DSSC 

In previous research, Tang et al. have reported the graphene based all-weather dye-sensitized solar cell, which harvests the energy not only from the sunlight, but also from rain [134,135,136,137]. The basic principle of all-weather solar cells is forming the π-electron|cation double-layer pseudocapacitance at graphene/raindrop interface. And then, the delocalized π-electron migrate along the penetrating pathways of raindrop forward at spreading process and backward at shrinking periods, finishing the charging and discharging process [135]. In conductive polymers, the conjugated structure between the heteroatom (N or S) and the benzene ring can form the graphene-like π-electron distribution system. In this sense, the conductive polymer can substitute the graphene in the all-weather dye-sensitized solar cell. At the interface between PANI and raindrop, the charge interactions occur, which can meanwhile harvest the energy from sunlight and rain [138]. The hybridized solar cell with PANI yield an efficiency of 6.5%. 

## 3. Polymers in Perovskite Solar Cells

The perovskite refers to the compounds with ABX_3_ structure, where A refers to a cation like formamidinium (FA^+^), methylammonium (MA^+^), or caesium (Cs^+^), B represents the metal cation like Pb^2+^ or Sn^2+^, and the X is halogen ions (F^−^, Cl^−^, and Br^−^). The organic-inorganic hybrid perovskite solar cell is triggering a revolution in the field of photovoltaic cells due to its booming performances [139,140,141,142]. The device architecture of PSCs can be divided into two distinct types: Regular n–i–p PSCs and inverted p–i–n PSCs. The schematic diagram of perovskite solar cell is shown in Figure 8 [143,144,145,146,147]: The ambipolar transmission characteristics of perovskite determine that it can generate electrons and holes after being excited by light. The electrons are excited to the CB of perovskite, while the holes left in the valence band (VB). The excited electrons in CB of perovskite transfer to the FTO substrate through the CB of the electron transport layer (ETL), and then flow through the external circuit. At the same time, holes in the VB of perovskite diffuse to perovskite/HTL interface, and then injected into the HTL and reached the metal electrode to complete the closed loop. The carriers recombination happens through the radiative or non-radiative transition. The back electron transfer at the interface of TiO_2_/perovskite and the back hole transfer at the interface of HTL/perovskite occur. The incomplete coverage could lead to the charge recombination at the interface of TiO_2_/HTL. Hence, the components materials, the crystallization morphologies of perovskite films, and the interfacial properties all will exert direct influences on device efficiency and stability.

### 3.1. Polymers in Perovskite Morphology Regulations

The intrinsic properties of organic-inorganic hybrid perovskite can be used to explain the rapidly increased efficiency, such as high absorption coefficient, direct and tunable bandgap, high carrier mobility and long carrier lifetimes [148,149]. The crystallization morphology of the perovskite film is another imperative consideration for high efficiency device, because it can exert the fatal influences on light absorption and carrier transportation dynamics [150]. Nevertheless, the rapid reaction between precursors introduce the ineluctable defects, which damage the device performances by accelerating the carriers recombination [151,152]. Therefore, more and more endeavors have been dedicated to reduce the defects and pinholes and to obtain the compact and even perovskite films. The additive engineering has been proved to be efficient in facilitating nucleation, regulating crystal growth, and enhancing device performances [153]. The existing additives include the follow aspects: (1) Solvent additives [153]; (2) fullerene additives [154]; (3) metal halide salt additives [155,156]; (4) inorganic acid additives [157,158]; (5) organic halide salt additives [157,158]; (6) nanoparticles additives [159]; (7) polymer additives [160,161]; and (8) other additives [162,163].

The obvious ionic nature of perovskite decides that perovskite can generate multiple chemical interaction with different functional groups [161]: The empty orbitals from Pb or Sn make it possible to form the coordination interaction with molecules containing hetero atoms [164]; The H atom in MA^+^ or FA^+^ can form hydrogen bonds with atoms with high electronegativity and small radius, such as “F, O, and N atoms” [165]. Polymers are a kind of macromolecule organics with various functional groups and hetero atoms [166]. Therefore, the polymers can be used as the additives to enhance the interaction between grains in perovskite films, enhancing the device stability further [165]. Besides, the good solubility of polymers in polar solvent reduces the contact angle, which guarantees the spread of precursor solution and the even coverage of perovskite film [167]. These unique features make the polymers can be used as additives to improve the crystallization morphology and device performances. Figure 9 shows the molecular structures of common polymers additives, involving poly(ethyleneglycol) (PEG), polyetherimide (PEI), polyvinylpyrrolidone (PVP), polystyrene (PS), poly(methyl methacrylate) (PMMA), 2,4-dimethyl-poly(triarylamine) (PTAA), Poly(2-ethyl-2-oxazoline) (PEOXA), poly[2-methoxy-5-(2-ethylhexyloxy)-1,4-phenylenevinylene] (MEH-PPV), poly-acrylonitrile (PAN), and poly[(9,9-bis(3’-(*N*,*N*-dimethylamino)-propyl)-2,7-fluorene)-alt-2,7-(9,9-dioctyl-fluorene)] (PFN), etc. [164,165,166,167,168,169,170,171,172,173,174,175]. Figure 10 presents the influences of several polymer additives on the surficial and cross-sectional morphology of perovskite films. It is obvious that theses polymer additives improve the coverage and reduce the pinholes, and are therefore conducive to reduce recombination and enhance device performance.

Masi et al. investigated the effects of following several polymers on the crystallization morphology of perovskite films systematically, such as MEH-PPV, PFN, PMMA, PS, and PTAA [157]. The results revealed that the films based on NEH-PPV and PFN additives are helpful to obtain the smooth morphology with small and uniformly distributed domains. The PMMA, PS, and PTAA based films present obvious bundle/island-like morphology, which can be attributed to the phase separations caused by high aggregation state. Chang et al. fabricated CH_3_NH_3_PbI_3-x_Cl_x_ perovskite with improved coverage by incorporating the PEG additive into the precursor solution [170]. The PEG is beneficial to spread out the precursor solution, to retard the crystallization rate, and to reduce the pinholes in grain boundaries [170]. The perovskite film based on 1 wt % PEG obtained an enhanced efficiency of 13.20%. Apart from the roles of improving the coverage and slowing down the crystallization rate, Zhao et al. found that PEG also served as the scaffold for perovskite film due to its three-dimensional network structure (Figure 11a–d) [171]. The device based on PEG scaffold structure retained 65% of its initial efficiency after aging for 300 h in highly humid condition with RH of 70%. Besides, the color of perovskite films and the device performances based the PEG scaffold can be self-healed quickly after removing the vapor spray (Figure 11e) [171]. The enhanced stability and self-healing effect of the PEG scaffold-based device results from that the hydrogen bonding interaction between perovskite and PEG and the excellent hygroscopicity of the PEG. Guo et al. fabricated the semitransparent solar cells assembled with stable cubic phase perovskite film by utilizing the PVP additive [172]. The PVP additive can enhance device efficiency and stability by reducing surface roughness, increasing shunting resistance, forming hydrogen bonds, and improving moisture resistance. Ultimately, 3.0 wt % PVP-doped perovkite solar cells obtain the enhanced efficiency than that of reference devices. Bi et al. also fabricate the efficient and stable perovskite solar cells by using the PMMA as template to regular the nucleation and growth of perovskite films [173]. Owing to forming the intermediate adduct with PbI_2_, the PMMA can adjust the preferred orientation of nuclei to minimize the total Gibbs free energy. The PMMA template based-device obtained the best efficiency of 21.3%.

### 3.2. Polymers as Hole Transport Layers (HTLs)

The electron injection process from the CB of perovskite to the CB of TiO_2_ finishes within 200 fs. The hole injection process from the VB of perovskite to that of HTL occurs in 0.75 ps. The results indicated that the recombination risks at the interface of perovskite/HTL are higher than that of perovskite/ETL [176]. Therefore, looking for an efficient hole transfer material is imperative and significant to the rapid electron-hole separation and transmission, and excellent device performances. The commonly used materials for HTL include three categories: Organic small molecule HTL, inorganic HTL, and polymeric HTL [177]. 

The Spiro-OMeTAD is one of the most typical organic small molecule HTL. However, the device based on pure Spiro-OMeTAD HTL always presents poor performances, which can be attributed to its intrinsic low hole-mobility (4 × 10^−5^ cm^2^·V^−1^·S^−1^) [178,179,180]. The lithium bis(trifluoromethanesul-fonyl)imide (Li-TFSI) and tertbutylpyridine (*t*BP) can be used as the efficient additives to increase the conductivity of Spiro-OMeTAD by an order of magnitude [180,181,182]. However, the easy deliquescence of Li-TFSI decides the decayed device efficiency and poor environmental stability [141,183]. Besides, the long oxidization process and the cumbersome synthesis and purification processes also greatly increase the cost of Spiro-OMeTAD [183]. Above mentioned drawbacks all limit the long-term utilization of Spiro-OMeTAD HTL. The inorganic HTLs (such as NiMgLiO, CuSCN, and CuI) present the easy fabrication, high hole mobility, and excellent stability [184,185,186]. However, the solvent used for the deposition of inorganic HTLs might dissolve perovskite partially, damaging perovskite film structure and compromising device performances further [142,176,187,188]. Compared with the organic small molecule HTLs and inorganic HTLs, the conjugated polymer HTLs possess high stability and solution operability, which have been evident from the OPVs. Figure 12 and Table 6 present the common chemical structures of polymer HTLs and corresponding device performance parameters. The common polymer HTLs include the following several types:

#### 3.2.1. Triarylamine-Based Polymers as HTLs

The first PSC device with polymer HTL is based on the (poly-[bis(4-phenyl)(2,4,6-trimethylphenyl)amine] (PTAA). This groundbreaking work illustrates that the triarylamine-based molecular is one kind of the most effective hole transfer materials, which also motivates the research about the HTLs based on the triarylamine and corresponding derivatives. Heo et al. constituted the PTAA HTL based device and yielded the efficiency of 9.0%, even higher than that based on Spiro-OMeTAD (8.4%) [189]. The strong interaction between PTAA and perovskite and its high hole mobility (1 × 10^−2^~1 × 10^−3^ cm^2^·V^−1^·S^−1^) are the main reasons for excellent efficiency [189]. Even though the device efficiency based the PTAA HTL is higher than that based on Spiro-OMeTAD, the device efficiency of 9.0% is relatively low. However, Yang et al. fabricated the device with the structure of FTO/(bl/mp-TiO_2_)/perovskite/PTAA/Au and obtained the efficiency of 20.2%, which fully affirms the possibility and potential of PTAA as HTL [190]. Qin. et al. fabricated the 2,4-dimethoy-phenyl substituted triarylamine oligomer S197 and applied it as the HTL of PSC [191]. The excellent solubility, suitable energy level, high hole mobility equip the S197 as a feasible HTL. This point was evident from its corresponding device efficiency of 12.00% with illumination of 99.6 mW·cm^−2^, higher than that based PTAA (11.50%). Therefore, it is significant to design and synthesis the novel derivatives based on the triarylamine and to employ them as the hole transfer materials. 

#### 3.2.2. Conductive Polymers as HTLs

The conductive polymers have been extensively used as the hole transfer materials in PSCs due to their easy fabrication process, low cost and p-type semiconductor nature. PEDOT:PSS is one of the most successful HTLs, especially in the inverted planar PSCs [192]. The transparency, good film forming properties, high work function (~5.2 eV), and low processing temperature of PEDOT:PSS enable it to serve as the effective HTL [192,193], especially in the flexible PSCs. Chen et al. fabricated the device with ITO/PEDOT:PSS/FASnI_3_/Spiro-OMeTAD/Au architecture and obtained the efficiency of 7.05% [194]. Bai et al. compared the performances of PSCs based on several different HTLs. The results illustrate that the PEDOT:PSS based device obtained the lower efficiency than others, which can be explained by the higher working function of PEDOT:PSS than the counterparts [195]. Zhang et al. obtained the efficiency of 17.16% by using the device with ITO/PEDOT:PSS/perovskite/PCBM/Bphen/Al [196]. By controlling the morphologies of perovskite and regulating the interfaces, the PEDOT:PSS-based PSCs can obtain the excellent efficiencies, even higher than 18% [197]. However, the stability of PEDOT:PSS based device is inferior, which results from the hydrophilic and acidic characteristics of PEDOT:PSS [197,198,199,200]. The PEDOT:PSS could results in the corrosion of substrates, especially for the ITO substrate [198,199,200,201]. Moreover, the devices based on the PEDOT:PSS HTL always achieved the low *V_OC_* due to the unmatched work function between perovskite ionization potential and PEDOT:PSS [199,202]. In order to solve these problems, the Nafion was integrated with PEDOT:PSS to fabricate a novel HTL by Ma et al. [202]. The PEDOT:PSS:Nafion increased the transmittance, enhanced the conductivity and hole mobility, improved the *V_OC_* to 1.02 V by reducing the work function, and enhanced the environmental stability due to the hydrophobic nature of Nafion. Eventually, the PSC based on the Nafion-modified HTL yield the efficiency of 16.72%, 23.76% higher than that based on the pure PEDOT:PSS (13.51%). 

Besides, the PANI has been employed as the HTL in solid-state DSSCs. Inspired by this, Xiao et al. fabricated the PANI HTL with brachyplast structure by using two-step cyclic voltammetry approaches [203]. The PANI-based PSC delivers an efficiency of 7.34% and remains 91.42% of its initial efficiency after 1000 h. As an analogy, the electropolymerized PEDOT electrode was used to assemble the bifacial PSCs, where the PEDOT was used as HTL [204]. Ultimately, the PEDOT HTL-based bifacial PSC device achieved the front and rear efficiencies of 12.33% and 11.78%, respectively. Above all, research illustrates that the conductive polymers are one types of promising hole transfer materials.

#### 3.2.3. Poly-3-hexylthiophene Based Polymers and Composites as HTLs

Poly-3-hexylthiophene (P3HT) was widely used as the HTL due to its low cost and doping-free features. However, the initial efficiency (<7%) of P3HT HTL based PSC can be ascribed to that the flat P3HT molecular is easy to cause the substantial charge recombination due to the close contact between the thiophene units and perovskite [205]. Besides, the device based on the P3HT HTL obtains the low *V_OC_*, which can be attributed to the high occupied molecular orbital (HOMO) energy level of P3HT [206]. Zhu et al. synthesized diketopyrrolopyrrole-based copolymer (P) and employed it as a hole transporting layer of CH_3_NH_3_PbI_3_ based device [207]. Owing to the hole mobility of P copolymer (1.95 cm^2^·V^−1^·S^−1^) is higher than that of P3HT (3 × 10^−4^ cm^2^·V^−1^·S^−1^), the device-based P HTL obtained the efficiency of 10.80%, higher than that based P3HT HTL (6.62%). However, the device efficiency based on the P3HT is always lower than that based Li-doped Spiro-OMeTAD, which can be ascribed to the low hole mobility of P3HT [208].

Many endeavors have been devoted to improve the hole mobility of P3HT, such as adding the additives or fabricating the composites of P3HT/carbon materials [208,209,210,211]. Li-TFSI and *t*BP co-doped P3HT, with an enhanced conductivity, improved the device efficiency from pristine 5.7% to 13.7% [208]. The bamboo-like carbon nanotube [209], graphdiyne [210], and SWCNT [211] were used to fabricate the composites with P3HT to improve the hole mobility. The device based on the P3HT/SWCNT-PMMA HTL yielded the efficiency of 15.3% and presented enhanced thermal stability and humidity stability (Figure 13) [211]. Looking for another derivatives or composites based on P3HT with high hole mobility is imperative.

#### 3.2.4. Other Polymer as HTLs

Heo et al. integrated the several HTLs into PSC devices and studied their effects on the device peformances, such as PTAA, P3HT, poly-[2,1,3-benzothiadiazole-4,7-diyl-[4,4-bis(2-ethyhex-yl)-4H- cyclopenta[2,1-b:3,4-b’]dithiophene-2,6-diyl]] (PCPDTBT), and poly-[[9-(1-octylnonyl)-9H-carbazole

-2,7-diyl]-2,5-thiophenediyl-2,1,3-benzothiadiazole-4,7-diyl-2,5-thiophenediyl]) (PCDTBT). The results found that the devices based on the PCPDTBT, PCDTBT, and P3HT HTLs show the efficiency of 5.3%, 4.2% and 6.7%, respectively, lower than that based on the PTAA (9.0%). The lower device efficiencies based on the PCPDTBT, PCDTBT, and P3HT HTLs can be attributed to their lower hole mobility (~1 × 10^−4^ cm^2^·V^−1^·S^−1^) than that of PTAA (1 × 10^−2^~1 × 10^−3^ cm^2^·V^−1^·S^−1^) [189]. Kwon et al. synthesized the Poly[2,5-bis(2-decyldodecyl)pyrrolo[3,4-c]pyrrole-1,4(2H,5H)-dione- (E)-1,2-di(2,2’-bithiophen-5-yl)et-hene] (PDPPDBTE) and incorporated it as the HTL, yielding an efficiency of 9.20%, higher than that of Spiro-OMeTAD (7.6%) [212]. The higher device efficiency based on the PDPPDBTE HTL can be ascribed to the higher hole mobility (0.32 cm^2^·V^−1^·S^−1^) than that of Spiro-OMeTAD (~10^−4^ cm^2^·V^−1^·S^−1^). After aging 1000 h under a 20% humidity atmosphere, the PDPPDBTE HTL based device remains 91.3% of its initial efficiency [212]. Kim et al. fabricated a novel polymeric HTM RCP based on the benzo[1,2-b:4,5:b’]dithiophene (BDT) and 2,1,3-benzothiadiazole (BT) [213]. As comparision, the P-OR, P-R, and Spiro-OMeTAD were also used as the HTLs. The PSC device based on the RCP HTL obtains the efficiency of 17.3%, higher than that based P-OR (2.6%), P-R (1.9%), and Spiro-OMeTAD (3.8%). The device efficiency can be explained from the highest hole mobility of RCP (3.09 × 10^−3^ cm^2^·V^−1^·S^−1^). After adding the Li-TFSI and *t*BP additives (ADDs), the device based on the HTL obtained the reduced efficiency of 16.5%. The device based on the RCP almost remains its pristine efficiency after aging 1400 h at 75% humidity. This research broadens our horizons of designing and selecting the novel hole transfer materials [213].

### 3.3. Polymers as Electron Transport Layers (ETLs)

In perovskite solar cells, the ETL plays the roles of collecting electron from perovskite films and transporting it into the external circuit. Therefore, an ideal ETL material should have high electron mobilities and matched energy level with perovskite. In regular n-i-p architecture, the common ETL materials are TiO_2_ and ZnO. In inverted p-i-n architecture, fullerene and corresponding derivatives have been used as the most common ETL materials. However, the temporal stability of fullerene and corresponding derivatives is also poor, which can be evident from the morphological changes upon recrystallization [214]. Therefore, the research about ETL is still in a beginning stage, especially for the non-fullerene ETL. In recent years, the organic polymers have been widely used as the electron transfer materials due to their tunable bandgap, excellent film-forming ability, good electron mobility, and low fabrication cost (Figure 13). The existing organic polymer ETLs are mainly based on the naphthalene diimide (NDI) and perylene diimide (PDI) cores. Naphthalene diimide (NDI) was chosen as it is one of the most explored electron deficient building block for n-type polymers. About NDI-based polymers ETL, Sun et al. developed the PFN-2TNDI ETL by introducing amine functional group to the side chains of fluorine unite of PF-2TNDI. The device assembled with PFN-2TNDI ETL exhibited the PCE of 16.7%, higher than that of PCBM based device (12.9%). The increased device performances comes down to two reasons: The amines on polymer side chains can passivate the surface traps of perovskite; the amines can reduce the work function of metal cathode by forming dipoles at interfaces [215]. Wang et al. have employed poly{[*N*,*N*’-bis(2-octyldodecyl)-1,4,5,8-naphthalenediimide-2,6-diyl]-alt-5,5’(2,2’-bithiophene)}(N2200) as the ETL of perovskite and obtained decent PCE of 8.15%, which is competitive to that of PCBM ETL (8.51%). Moreover, in order to prove the universality of organic polymers as ETL in perovskite solar cells, other polymers PNVT-8 and PNDI2OD-TT have also been tested and achieve the efficiency of 7.13% and 6.11%, respectively [216]. Kim et al. developed another P(NDI2DT-TTCN) copolymer ETL with NDI and dicyano-terthiophene. With the intramolecular interaction between the S–N groups, the dicyano-terthiophene shows a pseudo-planar structure, which decides the good electron transport ability and deep LUMO level of P(NDI2DT-TTCN). The device based on P(NDI2DT-TTCN) showed the PCE of 17.0%, outperforming than that of PCBM ETL-based device (14.3%). This study also illustrates that P(NDI2DT-TTCN) ETL is hydrophobic and can passivate the surface against ambient conditions. The P(NDI2DT-TTCN) ETL also exhibited better mechanical stability than PCBM-based ETLs in bending cycles, which indicates a better prospect for flexible PSC [217]. Guo et al. designed a series of PDI-based polymer PX-PDIs ETL with different X copolymerized units: Vinylene (V), thiophene (T), selenophene (Se), dibenzosilole (DBS), and cyclopentadithiophene (CPDT). The device based on PX-PDI ETL shows the highest PCE of 10.14%, which can be attributed to that PV-PDI ETL shows the deeper LUMO energy level and better planar structure than other PX-PDI (X = T, Se, DBS, and CPDT) [218]. All this research provide valuable design guidelines for devising new polymeric ETLs.

### 3.4. Polymer as the Interlayer

The perovskite solar cells present the layer-by-layer structure. The invert p-i-n architecture includes the FTO/HTL, HTL/perovskite, perovskite grain/grain, perovskite/ETL, ETL/Au, and Au/atmosphere interfaces. Charge extraction and separation occur at the interfaces, which could suffer from the recombination due to any possible interfacial defects. Therefore, interfacial chemical interaction is decisive factor for the device performances. The interface engineering has been used as a common method to regulate the morphologies of perovskite films, reduce the defects, and improve the device performances. The polymer interfacial modification has been regarded as one of the most effective methods of interface engineering (Figure 14). At the HTL/perovskite interface, Malinkiewicz et al. introduce the poly(*N*,*N*’-bis(4-butylphenyl)-*N*,*N*’-bis(phenyl)benzidine) (polyTPD) layer, which can transfer hole and block electron due to its LUMO is closer to the conductive bandgap energy [219]. Lin et al. introduced the DPP-DTT, PCDTBT, P3HT, and PCPDTBT layers at the PEDOT:PSS/perovskite interface. The perovskite films grown on PCDTBT shows the highest crystallinity intensity and device efficiency than others due to its matched energy level [220]. Wen et al. used the insulating PS to treat the initial perovskite films and formed the tunneling junction between the perovskite film and hole transfer film. The PS films is capable of accelerating the separation of photo-generated electrons and holes by selectively transfer hole while blocking electrons, improving the device efficiency from 15.90% to 17.80% [221]. At the interfaces between perovskite grains, Wang et al. used the chlorobenezene solution of PMMA to treat the as-prepared perovskite films, which can passivate the surface trap states in perovskite films and suppress the recombination [222]. It is worth stressing that manipulating the perovskite films with PMMA is beneficial to enhance the device stability by protecting the perovskite films from oxygen and moisture. At the perovskite/ETL interface, Wang et al. chose the PS, Teflon, and polyvinylidene-trifluoroethylene copolymer (PVDF-TrFE) as the tunneling materials. The tunneling materials allow electron transfer from perovskite to C60 ETL and blocks the holes. The device based on the PS interlayer given the highest PCE of 20.3%, higher than that of control devices (16.9%) [223]. At the ETL/metal interface, polyethyleneimine (PEIE) and poly[3-(6-trimethylammoniumhexyl) thiophene] (P_3_TMAHT) create surface dipole. The durface dipole make the work function of Ag decrease from original 4.70 to 3.97 eV (PEIE) and 4.13 eV (P_3_TMAHT), and the corresponding PCE increase from original 8.53% to 12.01% (PEIE) and 11.28 (P_3_TMAHT), respectively [224]. These studies provide a guideline for using the polymers to realize the interface regulation and modification.

## 4. Polymers in Organic Photovoltaics

The OPVs have attracted much attention since the 1950s. The first breakthrough of OPVs is the development from the initial single-layer architectures to donor-acceptor (D-A) bilayer architectures [11,225,226,227]. The second breakthrough of OPVs is the development from the planar heterojunction (PHJ) to bulk heterojunction (BJH) [228]. Compared to the PHJ, the BHJ configuration can improve the charge separation greatly, because the bicontinuous networks ensure the increased D/A interface, where the charge separation happens effectively [11,226,228]. The OPV is composed of the sandwich-like ITO anode/organic polymer active layer (D-A)/metal cathode. As shown in Figure 15, the organic polymer active layer produces the excitons under the excitation of light; The excitons diffuse in the bulk of the active layer; Once reaching to the interface of donor/acceptor, the excitons were separated into the electrons and holes, which were then transferred to the two electrodes to produce the photocurrent [228,229,230,231]. Therefore, the design and synthesis of D-A based active layers are imperative, but significant for the excitons separation and device performance. The traditional acceptors include the fullerene and its derivatives due to high electron affinity, long electron diffusion length, and rapid charge separation [11,232], such as C_60_, PC_60_BM, and PC_70_BM. However, the selection range of polymer donors is wider than that of acceptors. The ideal polymer donors should satisfy the following conditions: With the medium or low bandgap; with high carriers mobility; with matched energy levels; with good solubility [233,234,235,236]. Figure 16 and Table 7 show the molecular structures of selected polymer donor materials and the photovoltaic performances parameters of corresponding devices.

### 4.1. Polymers in Binary OPVs

#### 4.1.1. Polymers with Wide Bandgap as the Donor Materials of OPVs

The poly(2-methoxy-5-(2’-ethyl-hexyloxy)-1,4-phenylene vinylene) (MEH-PPV) is the first studied polymer donor. Yu et al. blended the MEH-PPV with PCBM as the active layer of OPV, which presents the device efficiency of 2.9% [237]. However, the low charge mobility and narrow spectral absorption range doubtlessly limit the wide application of MEH-PPV donor. 

Poly (thiophene)-based conjugated polymer represented by P3HT has been extensively used as the donor in OPV due to its high carrier mobility, good solubility, and excellent crystallinity and self-assembly performance [238]. Ma et al. fabricated the device with the ITO/PEDOT/P3HT:PCBM/Al architecture and obtained the efficiency of 5.0% [239]. Zhao et al. reported the OPV with the structure of ITO/PEDOT:PSS/P3HT:ICBA(1:1, *w*:*w*)/Ca/Al, achieving the efficiency of 6.5% [240], where the ICBA refers to the Indene-C60. After using the K^+^(PFCn6:K^+^) as the electron transfer layer (ETL), the device efficiency of ITO/ PEDOT:PSS/P3HT:ICBA/ETL/Ca/Al was enhanced to 7.5% [241]. However, the low efficiencies of P3HT-based devices result from that the large bandgap and energy level arrangement of P3HT affect the light-harvesting and *V_OC_*.

Besides, polyfluorenes is also one of the typical building blocks for donor due to its excellent features [242,243,244]: Polyfluorenes possess good charge transport performance; Polyfluorenes is facile to self-organize as the anisotropic liquid crystalline structure; Polyfluorenes can act as the electron or hole conductors; The rigid plane structure of polyfluorene equips it low HOMO energy level. Even though the extensive donors in PV are based on the polyfluorenes, the device efficiencies are still low, which can be attributed to its wide bandgap limits the spectral absorption range [245]. Therefore, much research concentrates on developing the polyfluorene copolymers with narrow bandgap and extended absorption [246,247]. Sevensson fabricated poly(2,7-(9-(2’-ethylhexyl)-9-hexyl-fluorene)-alt-5,5-(4’,7’-di-2-thienyl-2’,1’,3’-benzothiadiazole)) (PFDTBT), which presents the wide absorption and enhanced efficiency of 2.2 % in OPV [244]. The low photocurrent is the major barrier for achieving high efficiencies in this case. In order to solve this problem, Chen and Hou et al. grafted the different side chains on the PFDTBT build block to synthesize poly{[2,7-(9,9-bis-(3,7-dimethyl-octyl)-fluorene)]-alt-[5,5-(4,7-di-20-thienyl-2,1,3-Benzoth-adiazole)]} (BisDMO-PFDTBT) andpoly{[2,7-(9,9-bis-(2-ethylhexyl)-fluorene)]-alt-[5,5-(4,7-di-20-thienyl-10-2,1,3-benzothiadiazole)]} (BisEH-PFDTBT). The side chain modified PFDTBT yielded the highest efficiency of 4.5% by influencing the solubility, π–π stacking of polymers, and bandgap [248]. 

The donors based the MEH-PPV, P3HT, and polyfluorenes all possess the wide bandgap, which decides the low short current in corresponding devices. Hence, developing the donors with the medium or low bandgap is significant for improving the device efficiency of OPV.

#### 4.1.2. Polymers with Medium Bandgap as the Donor Materials of OPV

The thieo[3,4-b]thiophene-based structures have raised much attentions due to low bandgaps and improved device efficiency of 7~9% [11,249]. The polymers based on the benzodithiophene and thieno[3,4-b]thiophene (PTB7) units were widely used in the OPV and exhibited the excellent properties due to the synergistic roles [250,251]: The quinoidal structure unit results in a narrow bandgap of 1.6 eV; The plane conjugate structure leads to the good hole mobility; The related side chains enable the good solubility; The introduce of fluorine decreases the HOMO energy level, enhancing the *V_OC_*. All these advantages make the OPV device with the PTB7/PC_71_BM active layer (1:1.5, weight ratio) obtained the best efficiency of 6.22%, where the PTB7/PC_71_BM were dissolved in dichlorobenzene. When using the mixed solvent of chlorobenzene and 1,8-diiodoctane (97%:3% by volume), the morphology becomes even, and the efficiency was enhanced to 7.40% [251]. Chen and Hou et al. grafted the alkyloxy chain and alky side chain on the carbonyl of the thieno[3,4-b] thiophene unit of PBDTTT to synthesis the PBDTTT-E and PBDTTT-C, respectively. The PBDTTT-C was modified with a fluorine atom to form the PBDTTT-CF. The narrow bandgaps of PBDTTT-C, PBDTTT-E, and PBDTTT-CF (about 1.77 eV) ensure the wide spectrum absorption range, leading to the high short current density of 13~15 mA cm^−2^. Being influenced by the side chain and fluorine atom, the HOMO level of PBDTTT-CF was reduced, which resulted in the high *V_OC_*. Consequently, the corresponding device efficiency was enhanced to 7.73% [252].

He et al. adjusted the work function of ITO from 4.7 eV to 4.1 eV by introducing the PFN thin film on the top of ITO (Figure 16). The ohmic contact between modified ITO and photoactive layer was formed, which accelerates the transport and collection of the carriers. As a result, the device with inverted ITO/PFN/PTB7:PC_71_BM/MoO_3_/Al/Ag structure achieved the efficiency of 9.2% [253]. Except the above mentioned electron donors, the polymers based on the indacenodithiophene (IDT), indacenodithieno[3,2-b]thiophene (IDTT), and 1,3-bis(thiophen-2-yl)-5,7-bisethylhexyl) benzo[1,2-c:4,5-c’]dithiophene-4,8-dione (BDD) also have been used as the donor materials with medium bandgap due to their conjugate structure is helpful to improve the hole mobility [254,255,256,257].

#### 4.1.3. Polymers with Narrow Bandgap as the Donor Materials of OPV

Generally, the spectral absorption range of PCBM is always concentrated on 350–550 nm. The absorption edge wavelengths of polymers with wide or medium bandgaps are located at as far as 700 nm. In order to broad the light-harvesting range, more research is devoted to the design the electrons donors with narrow bandgap [257]. The benzo[1,2-b:4,5-b’]dithiophene (BDT) and diketopyrrolopyrrole (DPP) are two representative units for synthesizing the polymer donor materials with narrow bandgap. 

The large planar conjugated structure equips the BDT-based conjugated polymers with the large hole mobility. The small steric hindrance minimizes the impacts of adjacent units on the energy level and bandgap of BDT-based conjugated polymers. These two merits give the priority to use BDT unit to design the ideal donor materials [258]. The BDT unit has been recognized in the OPVs field since the Hou et al. synthesized the eight BDT-based copolymers and explored their photovoltaic properties [258]. The BDT can react with other conjugated units to form the photovoltaic active conjugated polymers, such as the thieno[3,4-b]thiophene (TT) [259], *N*-alkylthieno[3,4-c]pyrrole-4,6-dione (TPD) [260], and bithiazole [261], etc. Cui et al. reported that single-junction OPV baded on PBDB-T:ITCC-M and PBDTTT-E-T:IEICO active layers obtained the efficiency of 10.1% and 8.45%, respectively. The tandem OPV based on above two active layers yielded the best efficiency of 13.8% [262]. Zhao et al. fabricated the poly[(2,6-(4,8-bis(5-(2-ethylhexyl)thiophen-2-yl)-benzo[1,2-b:4,5-b’]dithiophene))-alt-(5,5-(1’,3’-di-2-thienyl-5’,7’-bis(2-ethy-hexyl)benzo[1’,2’-c:4’,5’-c’]dithiophene-4,8-dione))]) (PBDB-T) by modifying the BDT units [263]. The control device based on the PBDB-T:PC_71_BM yielded an efficiency of 7.45%. After alternating the PC_71_BM acceptor to 6-3,9-bis(2-methylene-(3-(1,1-dicyanomethylene)-indanone))-5,5,11,11-tetra-kis(4-hexylphenyl)-dithieno[2,3-d:2’,3’-d’]-s-indaceno[1,2-b:5,6-b’]dithiophene) (ITIC) [263]. The OPV based on PBDB-T:ITIC active layer obtained the best efficiency of 11.21% [264]. Meng and Chen et al. recently reported the tandem solar cell based on the ITO/ZnO/PFN-Br/PBDB-T:F-M/M-PEDOT/ZnO/ PTB7-Th:O6T-4F:PC_71_BM/ MoO_x_/Ag architecture and presented the complementary spectrum and matched energy level (Figure 17). The device efficiency reaches to 17.36%, breaking the world record for the efficiency of OPV [264].

The DPP is used to construct the low-bandgap conjugated polymers due to its electron-deficient nature. Dou et al. synthesized a series of conjugated polymers with low bandgap by using the BDT and DPP units [265]: PBDTT-DPP, PBDTP-DPP, PBDTT-FDPP, and PBDTP-DPP. The single layer device and the tandem solar cells based on the PBDTT-DPP:PC_71_BM obtain the best efficiency of 6.6%, and 8.8%, respectively [265]. Jung et al. reported a novel conjugated polymers PDTTDPP with the bandgap of 1.39 eV using the dithieno[3,2-b:20,30-d]thiophene (DTT) and DPP units. The chloroform solution of PDTTDPP shows the absorption onset at 1015 nm. The absorption even extended to 1055 nm when reducing the thickness of film. The device based the ITO/PEDOT:PSS/PDTTDPP:PC_71_BM (1:1.5)/LiF/Al obtain the highest efficiency of 6.05% [266]. Hence, designing and synthesising the polymeric donor materials with narrow bandgap are the key to improve the light-harvesting and device efficiency of organic solar cells.

### 4.2. Polymers in Ternary OPVs

The traditional organic solar cells are composed of polymer donors and fullerene-based acceptors. Despite high electron mobilities, fullerene and its derivatives always show large band gap and narrow absorption window, which hinder the further development of OPVs. Therefore, a series of nonfullerene acceptor (NFA) materials have been reported due to its strong light absorption and easily tunable energy levels. Wang et al. introduce a non-fullerene small molecule m-ITIC into PTB7-Th:TQ1 system to fabricate the ternary OPV [267]. The m-ITIC can broaden the spectrum absorption range of OPV, enhancing the resultant device performances further. However, the electron mobilities of NFAs are always low, which affect the device performance directly. Hence, the fullerene acceptor was introduced into the binary polymers donor: Nonfullerene acceptor system to fabricate the ternary OPV. As the third component, the fullerene can compensate the deficiencies of nonfullerene acceptors. Fu et al. fabricated the ternary OPVs using wide-bandgap PTB1-C as donor and highly conductive PC_71_BM and narrow-bandgap nonfullerene IT-2F as acceptor [268]. The resultant device obtained increased PCE, which benefits from the increased light harvesting from of IT-2F and enhanced electron transport. Meanwhile, the addition of PC_71_BM can also suppress the aggregation of IT-2F, reducing the trap density and trap-assisted recombination. When it comes to the ternary OPVs, it is necessary to consider possible and unfavorable large-scale phase separation. Therefore, it is difficult, but imperative, to control the morphology while pursuing high performance. Xie et al. reported that ITIC-2Cl molecules could easily form granular aggregates in the binary blend [269]. After adding 20% ICBA into the blend, the aggregation of ITIC-2Cl reduced, which can be attributed to the amorphous ICBA can disturb intermolecular π–π interactions of ITIC-2Cl. The optimized morphology, low LUMO level, and complementary absorption decide an enhanced PCE of 13.4%.

### 4.3. Polymers as Buffer Layers of OPV

In organic photovoltaivs, there is a barrier between the active layer and the electrode. The barrier hinders the quick charge transfer, leading to serious charge accumulation at the interfaces. The charge accumulation increase the recombination probability, and attenuate the device performances. Therefore, it is imperative to accelerate the charge separate and transfer efficiency. In order to solve this problem, an interface buffer layer is introduced in the OPVs, including the anode buffer layer and cathode buffer layer. These buffer layers are often established between the active layer and the electrode to form an ohmic contact, which is conducive to the efficient charge extraction and separation. Besides, the buffer layer also can inhibit the possible diffusion and reaction between the active layer and the electrode material. The polymer serves as one kind of the most effective buffer layers. As Table 7 summarized, the PEDOT:PSS has been widely used as the most common anode buffer layer to extract holes, and the corresponding devices achieve enhanced efficiency. Despite the effective charge extraction ability, the acidic PEDOT:PSS solution will cause a certain degree of corrosion of ITO substrate, which in turn reduces the device stability [270]. For the cathode buffer layers, He et al. have applied the alcohol/water-soluble poly[(9,9-bis(3’-(*N*,*N-*diMethylaMino)propyl)-2,7-fluorene)-alt-2,7-(9,9-dioctylfluorene)] (PFN) to extract electron in organic photovoltaics [271]. The OPVs assembled with the PFN cathode buffer layers show enhanced *J_SC_*, *V_OC_*, and *FF*, which results in a PCE of 8.37% and 6.79% for PTB7 and PCDTBT devices. The positive effect of PFN buffer layer on device performances include: Forming an enhanced built-in potential across the device; and improving the charge separation and transfer efficiency. Na et al. use polyfluorene derivative (WPF-oxy-F) as the interfacial layer between the active layer and metal cathode, which will reduce the metal work function for better electron transport [272]. The PF-EP was introduced as a cathode buffer layer to improve the charge collection efficiency by preventing the metal atoms migrating into the active layer [273].

### 4.4. Polymer-Based Micro/Nanostructures in the OPVs

The power conversion efficiencies (PCE) of organic photovoltaics (OPV) is still limited and cannot be compared with their inorganic counterpart. The low light trapping efficiency and low charge carrier mobility are the main limitations. 

Enhancing light trapping in OPV is important. A thin active layer always leads to optical losses accounting for 40% of total losses [274]. In planar heterojunction structure, the common exciton diffusion length is around 5–10 nm, which decides that the thickness of the active layer is about 10 nm [275]. In bulk heterojunction structure, the thickness of the active layer (about 100~200 nm) is still not enough to absorb and trap light efficiently due to low charge carrier mobilities in most organic materials. Therefore, improving the light trapping within existing OPV architectures is imperative. Apart from developing new materials, another way of improving the light trapping include the uses of textured substrates, noble metal nanoparticles, tandem solar cells, and microlens arrays (MLA). In the case of textured substrates, the aspect-ratio of textures need to be designed optimally to ensure the conformal coating of the active layers [276]. The unreasonable design will result in shunts and recombination [276]. The noble metal nanoparticles doped active layers employ plasmonic near-field enhancement effects to trap more light. However, the excessive metallic and surface stabilizer on metal could serve as the recombination centers to retard charge transport. In terms of tandem polymer solar cells, the folded structure cause light trapping at high angles and large photocurrent density. Besides, the tandem polymer solar cells also allow multiple bandgap solar cells series or parallel connection. Tvingstedt et al. reported a tandem cell by folding two planar but different cells toward each other, where single cells reflect the non-absorbed light upon another adjacent cell to realize the spectral broadening and light trapping [277]. The ultimate device performances were enhanced. Designing MLA structures on the backside of the transparent conductive substrates has been demonstrated as a universal method to increase light trapping of OPV. This approach couldn’t exert influences on the device fabrication processing and active layer morphology. Chen et al. have designed the near-hemispherical MLA of 2 mm diameter, which shows the higher ability to refract incoming light than a pure-hemispherical shape [278]. The diameter of this MLA is close to visible light wavelength. This property decides that the near-hemispherical MLA can not only reduce surface reflections, but also can utilize optical interference to enhance light intensity inside the active layer. Ultimately, for the P3HT:PCBM active layer system, the device with near-hemispherical MLA obtained the increased absorption, absolute external quantum efficiency (EQE), and PCE by 4.3%. For the PCDTBT:PC_70_BM active layer system, near-hemispherical MLA increases the absorption, EQE and PCE by 10.0%. All of these enhancements can be attributed to the increased light path and absorption, as well as diffraction induced enhanced light intensity. Tvingstedt et al. also studied and demonstrated the principle of MLAs trap the light in detail [279]. The MLAs can make the light display strong directional asymmetric transmission, realizing the recycle of reflected photons. Peer et al. design the device based on dual photonic crystals, where polymer MLAs on the glass was coupled with a photonic-plasmonic crystal at the metal cathode on the back of the cell. The MLAs focuses light on the periodic nanostructure, realizing strong light diffraction. The surface plasmon and waveguiding effect of photonic-plasmonic crystal enhance long wavelength absorption. The joint efforts of MLAs and photonic-plasmonic crystal result in absorption enhancement of 49% and photocurrent enhancement of 58% [280]. In addition to the OPV, the MLAs have also been extended to improve the device performances of perovskite and organic light emitting diodes [281]. 

Improving the charge carrier mobility and reducing the recombination are imperative to enhance the device efficiency of OPV. For a common device with BJH structure, the electron donor and acceptor interspersed randomly in the active layer, which always leads to the discontinuous electron transport pathways. Hence, the internal interfacial area must be large enough to create the continuous pathways for charge separation and transport. In order to solve this problem, He et al. constitute the nanostructured polymer heterojunctions of composition and morphology in the active layer through a double nanoimprinting process [282]. The devices based on double-imprinted poly((9,9-dioctylfluorene)-2,7-diyl-alt-[4,7-bis(3-hexylthien-5-yl)-2,1,3-benzothia-diazole]-2’,2’’-diyl) (F8TBT)/poly(3-hexylthiophene) (P3HT) films obtained a PCE of 1.9%, higher than that of planar F8TBT/P3HT (0.36%). The enhanced device performance can be attributed to ordered structure in the active layer can enhance the separation of electron-hole pairs and improve carrier mobilities further. Wiedemann et al. fabricate similar PCBM/P3HT bilayer devices with controlled nanostructured interfaces by combining nanoimprinting and lamination techniques, which presented a higher PCE (0.05%) than that of common bilayer structured device (0.03%) [283]. This method is also suitable for any other polymer combinations. 

## 5. Summary and Outlook

The polymers have been widely used in the photovoltaic fields, including the DSSC, PSC, and OPV. 

In DSSC, the polymers not only can be used as the flexible substrates, but also the pore- and film- forming agents of photoanode films; Besides, the conductive polymers and corresponding composites can be used to fabricate the platinum-free counter electrode materials due to their high catalytic activity; The long chain structure and containing functional groups also decide the polymers can be used to solidify the liquid electrolyte and form the quasi-solid-state electrolytes. The applications of polymers in flexible PAN or PET substrates, mesoporous TiO_2_ photoanodes preparation, polymer gel electrolyte and platinum-free counter electrodes are extensive and mature. The conductivity of quasi-solid-state polymer electrolytes is still low. Hence, improving the conductivity of quasi-solid-state electrolytes is still challenging, which can be realized by adjusting the ratios of polymers and liquid electrolyte.

In PSC, the polymers can be used to facilitate the nucleation, regulate the crystallization of perovskite films, and enhance the device stability by forming various interactions with the perovskite films. The polymers can also be employed as the hole transfer materials due to their high hole mobility. It is challenging to design the novel polymers hole transport materials with high hole mobility and appropriate energy level arrangement. Applying the polymers to fabricate the long-term stable PSC devices is another bottleneck. However, these two aspects are not only imperative to improve the carrier separation efficiency and the final device efficiency, but also important for the practical application.

In OPV, the polymers are widely used as the active layers to influence the light harvesting efficiency and device performances. The key to realizing the high-efficient OPV is to design the novel polymer donors with narrow bandgap and appropriate energy level arrangement. Besides, it is promising to select the polymers with complementary spectrum absorption range to fabricate the ternary or tandem OPVs.

## Figures and Tables

**Figure 1 polymers-11-00143-f001:**
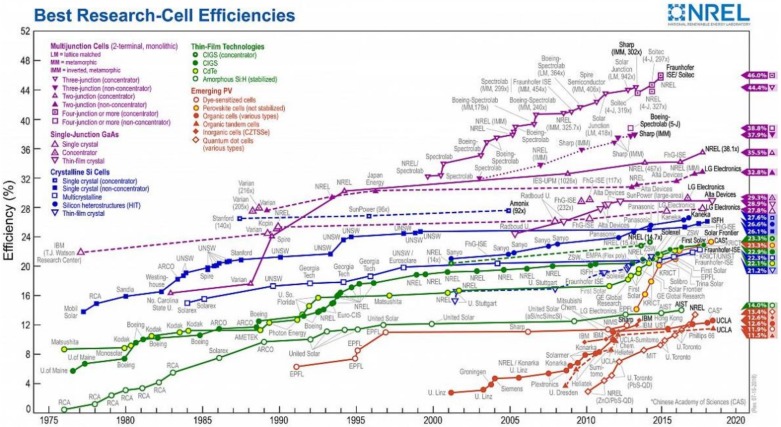
Photoelectric conversion efficiencies for various photovoltaic technologies since 1976 by National Renewable Energy Laboratory (NREL) [15].

**Figure 2 polymers-11-00143-f002:**
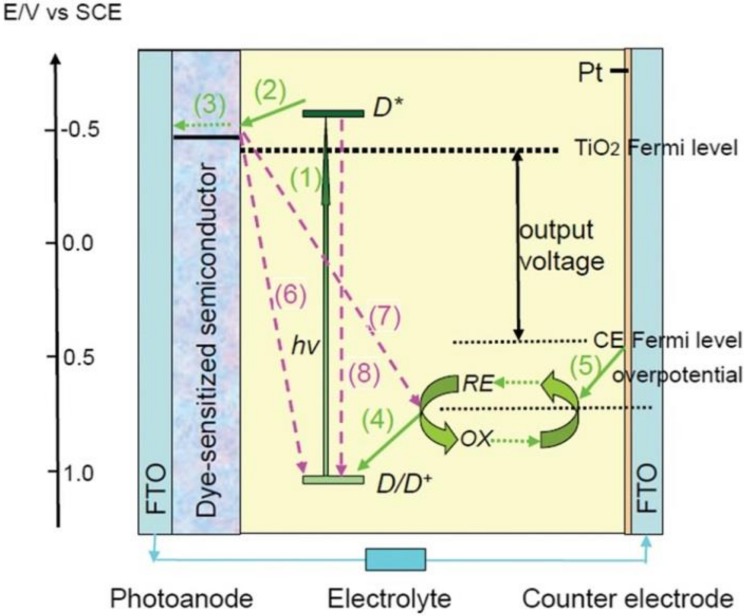
Fundamental processes of dye-sensitized solar cells (Reproduced from Reference [32] with permission, copyright The Royal Society of Chemistry, 2017).

**Figure 3 polymers-11-00143-f003:**
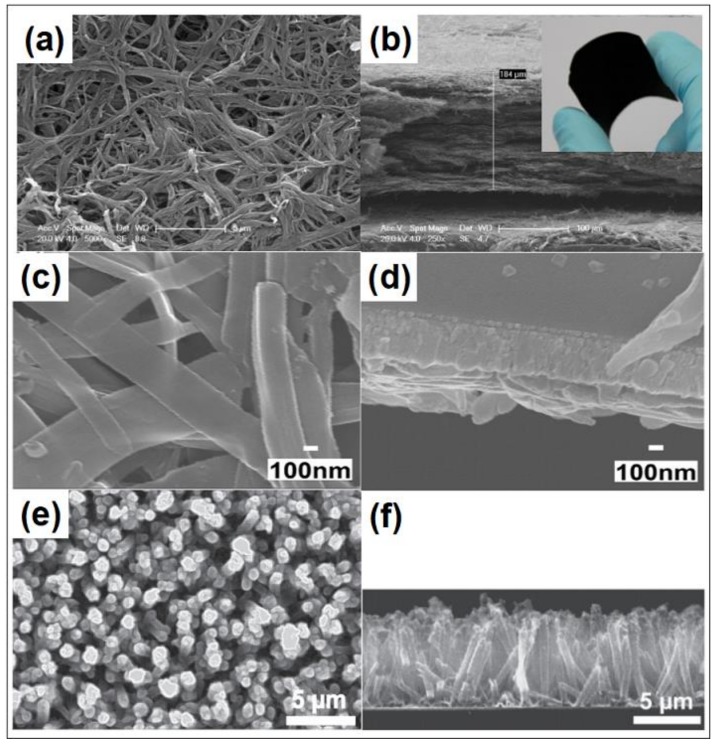
Field Emission Scanning Electron Microscope (FESEM) images and corresponding cross-sectional FESEM images of different conductive polymers counter electrodes: (**a**,**b**) Paper-like PPy membrane (reproduced from Reference [76] with permission, copyright American Chemical Society, 2014); (**c**,**d**) polyaniline (PANI) nanoribbons (reproduced from Reference [83] with permission, copyright Elsevier, 2016); and (**e**,**f**) PEDOT nanotubes (reproduced from Reference [93] with permission, copyright Wiley, 2011), respectively.

**Figure 4 polymers-11-00143-f004:**
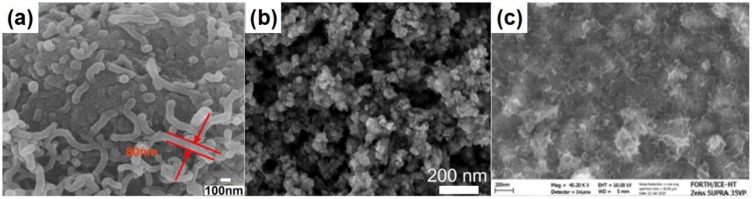
FESEM images of different hybrids counter electrodes based on conductive polymers: (**a**) PPy/MWCNT (reproduced from Reference [101] with permission, copyright Elsevier, 2016); (**b**) TiN particles-PEDOT:PSS (reproduced from Reference [102] with permission, copyright American Chemical Society, 2012); and (**c**) RGO/PPy/PEDOT composites (reproduced from Reference [104] with permission, copyright Elsevier, 2015), respectively.

**Figure 5 polymers-11-00143-f005:**
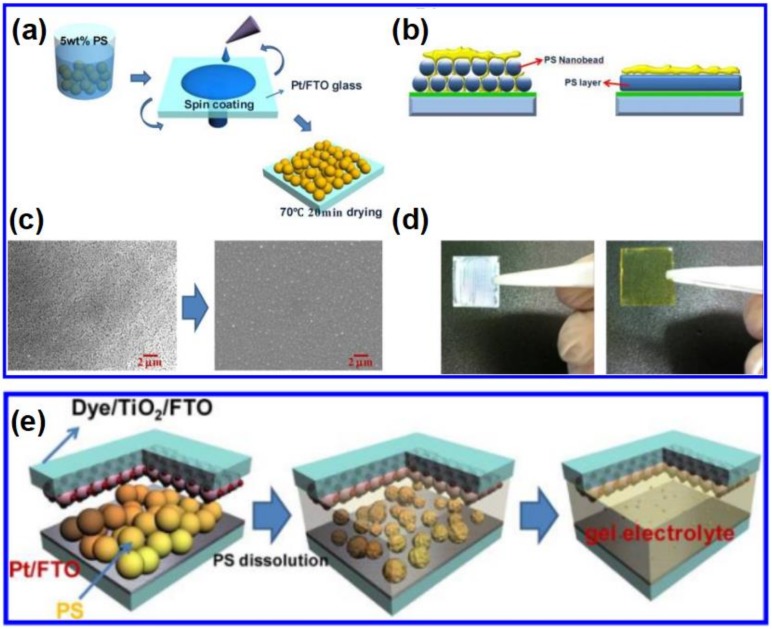
(**a**) Preparation schematic of PS nanobeads on FTO/Pt substrate; (**b**) Comparison schematic of the pore-filling before (left) and after (right) dissolving the PS nanobeads; (**c** and **d**) FESEM images and photographs of the PS nanobeads on the FTO/Pt substrate before (left) and after (right) the dissolution process; (**e**) Conversion schematic of liquid electrolyte to gel electrolyte by using valeronitrile solvent to dissolve the PS nanobeads (reproduced from Reference [123] with permission, copyright American Chemical Society, 2012).

**Figure 6 polymers-11-00143-f006:**
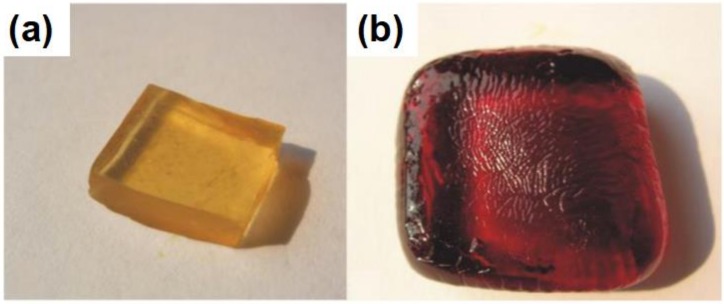
Photographs of thermosetting polymer electrolyte (PAA-PEG/NMP + GBL/NaI + I_2_): (**a**) Before and (**b**) after soaking in I_3_^−^/I^−^ liquid electrolyte, respectively (reproduced from Reference [129] with permission, copyright Wiley, 2007).

**Figure 7 polymers-11-00143-f007:**
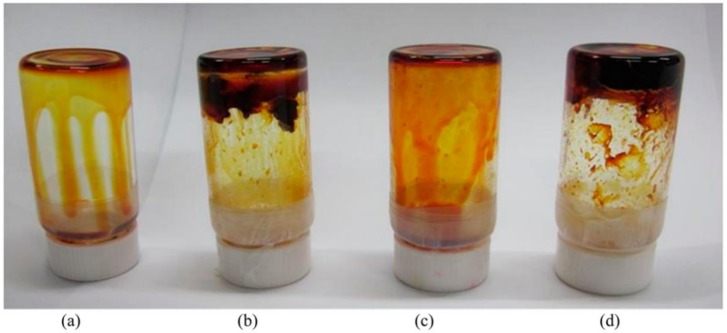
Upside-down vessel containing the PMII/I_2_/PEGDME electrolyte with (**a**) none, (**b**) 9 wt % of fumed silica, (**c**) 9 wt % of silica rods, and (**d**)12 wt % of silica rods, respectively (reproduced from Reference [132] with permission, copyright American Chemical Society, 2014).

**Figure 8 polymers-11-00143-f008:**
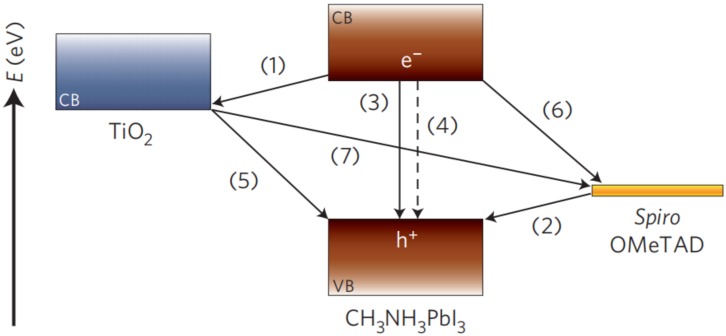
Schematic diagram of energy levels arrangement in perovskite solar cells (reproduced from Reference [145] with permission, copyright Nature Publishing Group, 2014).

**Figure 9 polymers-11-00143-f009:**
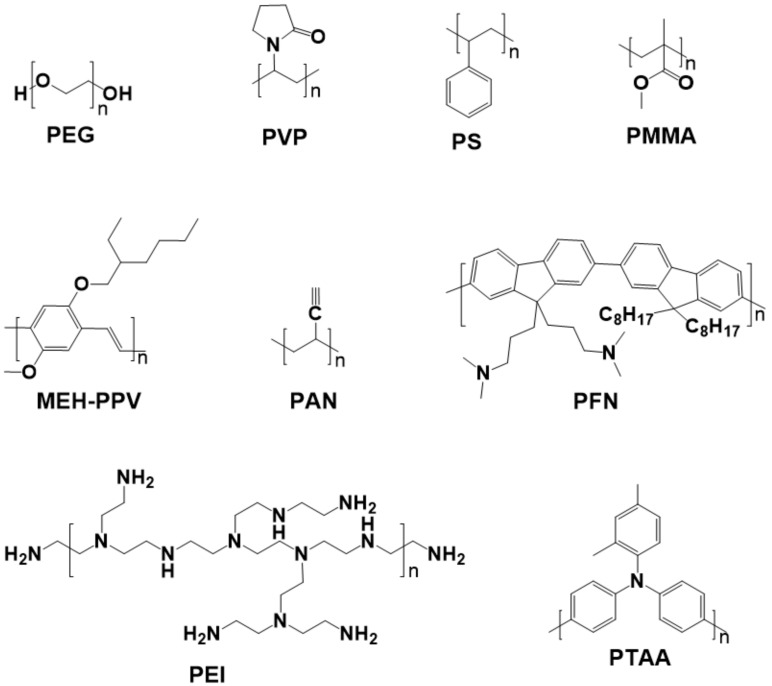
Molecular structures of common polymer additives.

**Figure 10 polymers-11-00143-f010:**
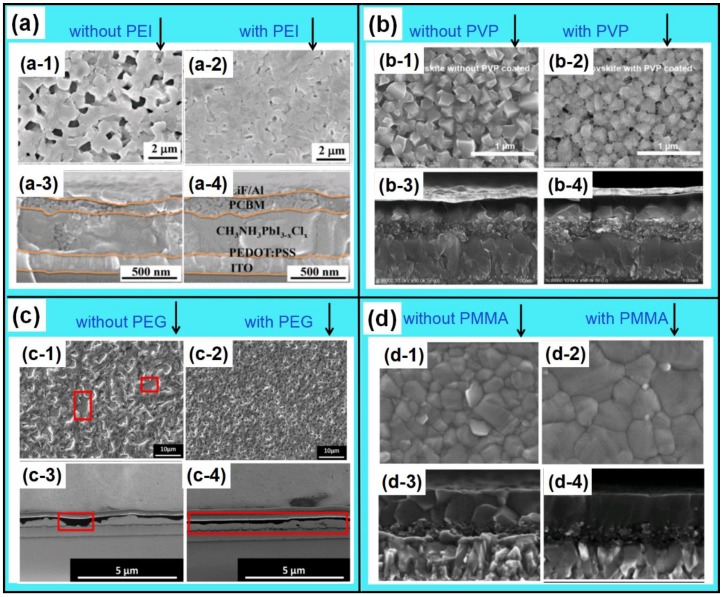
Comparisons of FESEM images and corresponding cross-sectional FESEM images of perovskite films with and without several different polymer additives: (**a**) PEI (reproduced from Reference [165] with permission, copyright The Royal Society of Chemistry, 2016), (**b**) PVP (reproduced from Reference [166] with permission, copyright Wiley, 2017), (**c**) PEG (reproduced from Reference [170] with permission, copyright American Chemical Society, 2015), and (**d**) PMMA (reproduced from Reference [173] with permission, copyright Nature Publishing Group, 2014), respectively.

**Figure 11 polymers-11-00143-f011:**
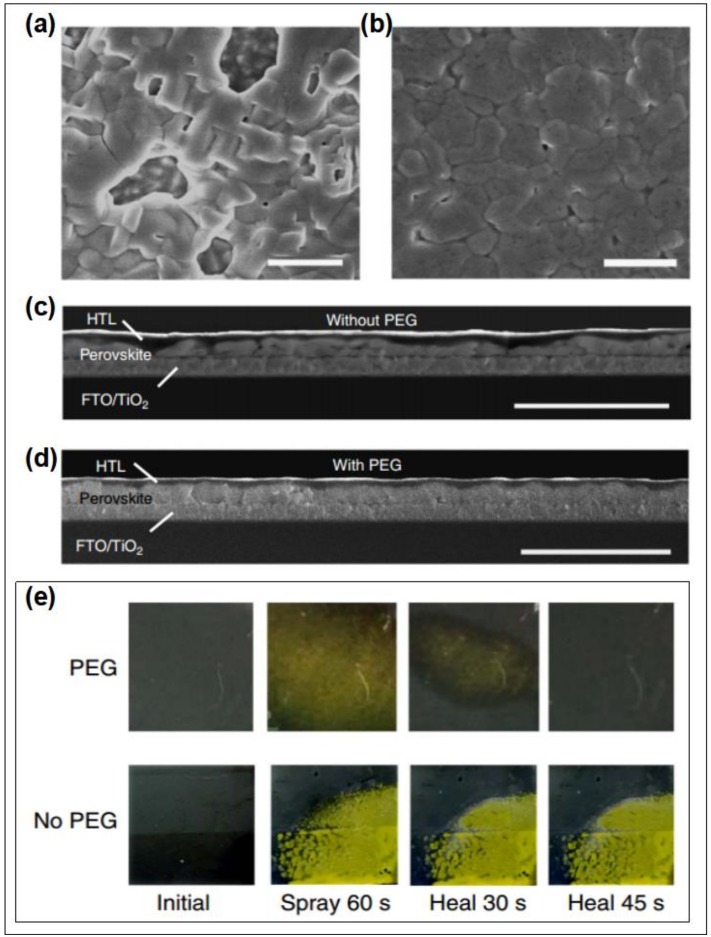
Comparisons of FESEM and corresponding cross-sectional FESEM images of perovskite films: (**a**,**c**) With PEG, and (**b**,**d**) without PEG, respectively. (**e**) Photographs of the color change of perovskite films with and without PEG after water-spraying for 60 s and kept in ambient air for 45 s (reproduced from Reference [171] with permission, copyright Nature Publishing Group, 2016).

**Figure 12 polymers-11-00143-f012:**
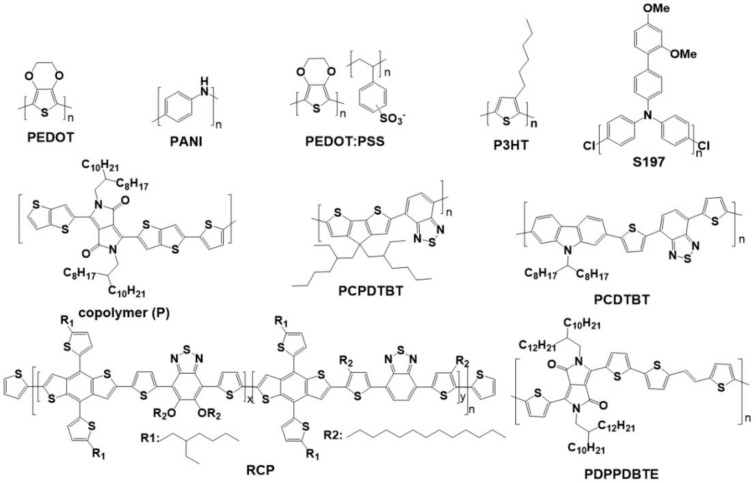
Chemical structures of polymers used to discuss the hole transport materials.

**Figure 13 polymers-11-00143-f013:**
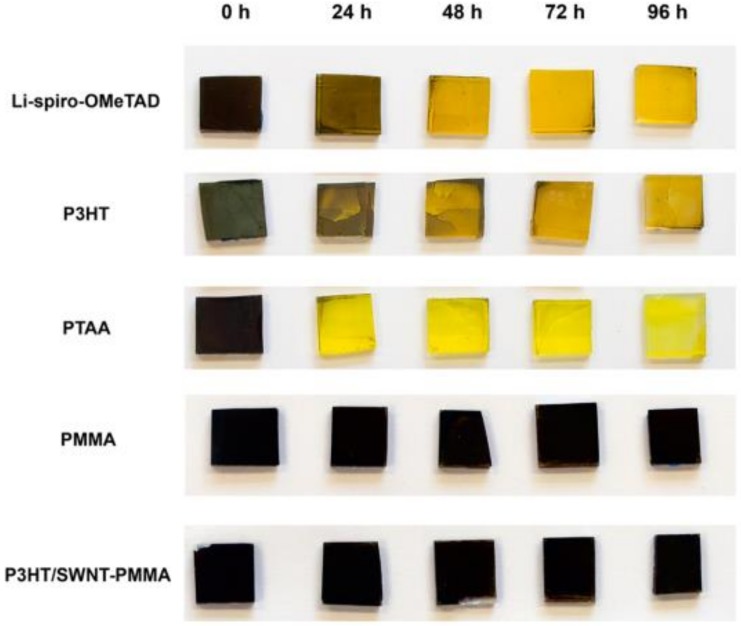
Visible degradation photographs of perovskite films covered with different hole transport materials, including the Li-spiro-OMeTAD, P3HT, PTAA, PMMA and P3HT/SWNT-PMMA (reproduced from Reference [211] with permission, copyright American Chemical Society, 2014).

**Figure 14 polymers-11-00143-f014:**
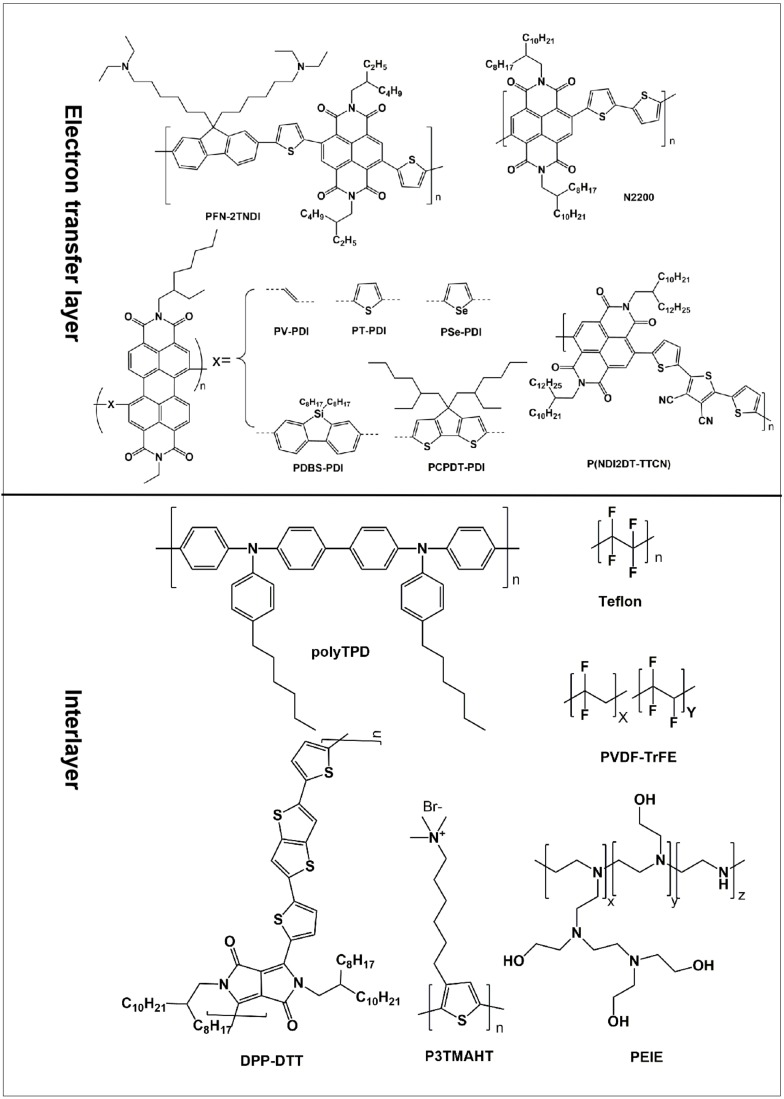
Chemical structures of polymers used to discuss the electron transport layers and interlayers.

**Figure 15 polymers-11-00143-f015:**
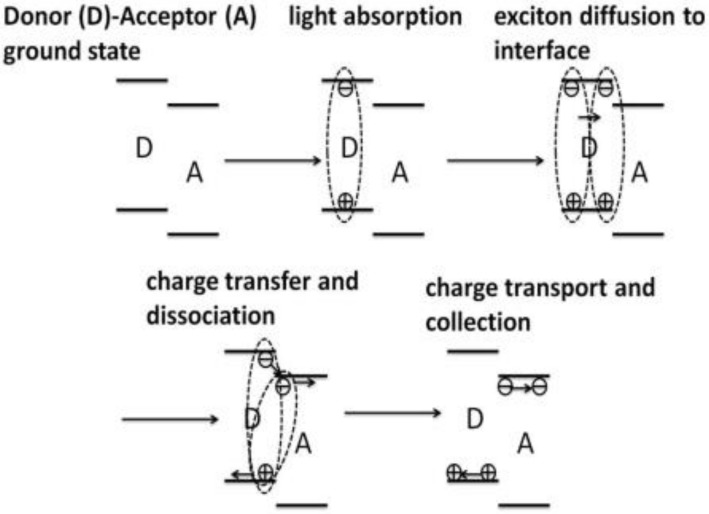
Mechanism diagram of organic photovoltaics (reproduced from Reference [230] with permission, copyright Wiley, 2014).

**Figure 16 polymers-11-00143-f016:**
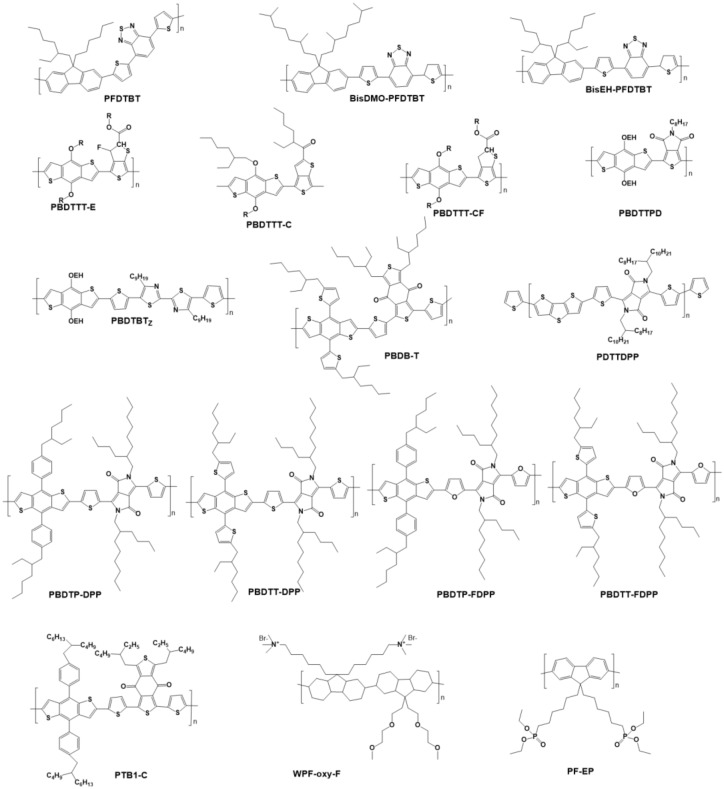
Molecular structures of materials for photoactive layers.

**Figure 17 polymers-11-00143-f017:**
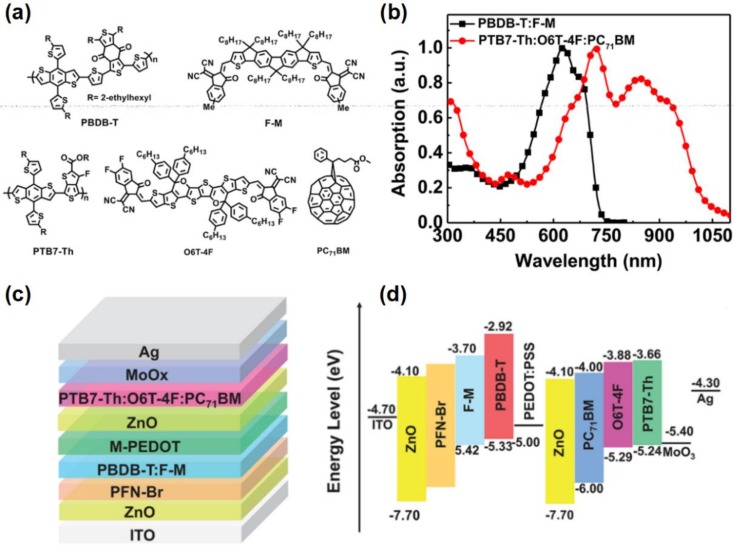
(**a**) Molecular structures of PBDB-T, F-M, PTB7-Th, O6T-4F and PC_71_BM; (**b**) Normalized absorption spectra of PBDB-T:F-M and PTB7-Th:O6T-4F:PC_71_BM films; (**c**) and (**d**) Device architecture and energy level diagram of the tandem cell, respectively (reproduced from Reference [264] with permission, copyright Science, 2018).

**Table 1 polymers-11-00143-t001:** Photovoltaic performances parameters of DSSCs based on the PPy counter electrodes.

Preparation Methods of PPy Counter Electrode	Electrolyte	*J_SC_* (mA cm^−2^)	*V_OC_* (V)	*FF* (%)	*η* (%)	Ref.
FeCl_3_ chemical polymerization	I3^−^/I^−^	13.80	0.771	26.3	2.86	[64]
Potentiostatical electrodeposition	I3^−^/I^−^	13.10	0.660	53.9	4.65	[64]
FeCl_3_ oxidation/Drop-casting	I3^−^/I^−^	15.50	0.791	67.0	8.20	[65]
FeCl_3_ oxidation	I3^−^/I^−^	13.96	0.712	64.9	6.45	[73]
FeCl_3_ oxidation/with HCl vapor post-doping	I3^−^/I^−^	11.90	0.725	63.2	5.70	[74]
FeCl_3_ oxidation/without HCl vapor post-doping	I3^−^/I^−^	15.20	0.721	61.8	6.80	[74]
Liquid-liquid biphasic interfacial polymerization	I3^−^/I^−^	11.31	0.49	63.0	3.50	[75]
FeCl_3_ oxidation/Heating pulp-like suspensions	I3^−^/I^−^	13.10	0.716	56.0	5.27	[76]
I_2_ oxidation	I3^−^/I^−^	15.01	0.74	69.0	7.66	[77]
APS oxidation	I3^−^/I^−^	12.19	0.725	52.0	5.74	[78]
V_2_O_5_ nanofibers/FeCl_3_ oxidation	I3^−^/I^−^	15.37	0.69	64.0	6.78	[79]

**Table 2 polymers-11-00143-t002:** Photovoltaic performance parameters of DSSCs based on the PANI counter electrodes.

Preparation Methods of PANI Counter Electrode	Electrolyte	*J_SC_* (mA cm^−2^)	*V_OC_* (V)	*FF* (%)	*η* (%)	Ref.
APS oxidation	I3^−^/I^−^	15.24	0.71	60.4	6.54	[63]
KPS oxidation	I3^−^/I^−^	13.50	0.76	38.3	3.92	[64]
cyclic voltammetry deposition	I3^−^/I^−^	13.40	0.728	67.6	6.58	[64]
APS oxidation	I3^−^/I^−^	14.60	0.714	69.0	7.15	[82]
V_2_O_5_ nanofibers oxidation/template etching	I3^−^/I^−^	17.92	0.72	56.0	7.23	[83]
In situ oxidation with fixed content of APS	Co(bpy)^33+/2+^	15.09	0.78	70.0	8.24	[84]
Drop-casting	Co(bpy)^33+/2+^	12.76	0.72	65.0	5.97	[84]
I-t electropolymerization	I3^−^/I^−^	10.88	0.69	58.0	4.35	[85]
Pulse electropolymerization	I3^−^/I^−^	12.18	0.71	60.0	5.19	[85]
Cyclic voltammetry deposition with SO_4_^2−^ doping	I3^−^/I^−^	10.70	0.81	64.0	5.60	[87]

**Table 3 polymers-11-00143-t003:** Photovoltaic performances parameters of DSSCs based on the PEDOT counter electrodes.

Preparation Methods of PEDOT Counter Electrodes	Electrolyte	*J_SC_* (mA cm^−2^)	*V_OC_* (V)	*FF* (%)	*η* (%)	Ref.
Fe(OTs)_3_ oxidative polymerization	I3^−^/I^−^	17.00	0.72	66.0	8.08	[69]
Constant current deposition	I3^−^/I^−^	13.96	0.7116	70.0	6.96	[70]
Potentiostatical electrodeposition	Disulfide/thiolate	15.90	0.687	72.0	7.90	[86]
Electrochemical polymerization	I3^−^/I^−^	8.84	0.705	63.0	3.93	[91]
cyclic voltammetry/ template etching	I3^−^/I^−^	16.75	0.75	64.0	8.05 (front)	[92]
cyclic voltammetry/ template etching	I3^−^/I^−^	7.84	0.719	67.0	3.78 (rear)	[92]
cyclic voltammetry electrodeposition	I3^−^/I^−^	17.72	0.768	67.0	9.12 (front)	[92]
cyclic voltammetry electrodeposition	I3^−^/I^−^	11.23	0.731	70.0	5.75 (rear)	[92]
ZnO nanowire template/electropolerization	I3^−^/I^−^	16.24	0.72	70.0	8.30	[93]
Electropolymerization with ClO_4_ doping	I3^−^/I^−^	9.60	0.68	66.0	4.20	[94]
Electropolymerization with TsO doping	I3^−^/I^−^	9.10	0.68	67.0	4.20	[94]
Electropolymerization with TsO doping	I3^−^/I^−^	9.20	0.665	66.0	4.0	[94]

**Table 4 polymers-11-00143-t004:** Photovoltaic performances parameters of DSSCs assembled with conductive polymers-based hybrids counter electrode.

Counter Electrodes	Preparation Methods	Electrolyte	*J_SC_* (mA·cm^−2^)	*V_OC_* (V)	*FF* (%)	*η* (%)	Ref.
PEDOT/*N*-doped GO	Constant current deposition	I_3_^−^/I^−^	15.60	0.7392	72.0	8.30	[70]
rGO@PPy	Electrochemical polymerization	I_3_^−^/I^−^	7.49	0.70	42.0	2.21	[71]
PPy/FeS	FeCl_3_ oxidation/ Na_2_S reduction	I_3_^−^/I^−^	15.87	0.711	66.3	7.48	[73]
PANI-graphene	Cyclic voltametric	I_3_^−^/I^−^	16.55	0.699	67.0	7.70	[96]
PANI-MWCNT	Cyclic voltametric	I_3_^−^/I^−^	22.25	0.691	60.1	9.24 (both)	[97]
PANI-MWCNT	Cyclic voltametric	I_3_^−^/I^−^	17.95	0.675	65.3	7.91 (front)	[97]
PANI-MWCNT	Cyclic voltametric	I_3_^−^/I^−^	4.30	0.622	64.3	1.72 (rear)	[97]
PPy-SWCNT	FeCl_3_ oxidation	I_3_^−^/I^−^	15.68	0.742	71.0	8.30	[98]
PANI-MWCNT	Pulse potentiostatic	I_3_^−^/I^−^	13.53	0.721	64.0	6.24	[99]
PEDOT/MWCNT	Cyclic voltametric/ PMMA template	I_3_^−^/I^−^	17.09	0.792	67.0	9.07 (front)	[100]
PEDOT/MWCNT	Cyclic voltametric/ PMMA template	I_3_^−^/I^−^	10.76	0.757	69.0	5.62 (rear)	[100]
TiN(P)-PEDOT:PSS	Doctor blade	I_3_^−^/I^−^	14.45	0.727	67.18	7.06	[102]
TiN(R)-PEDOT:PSS	Doctor blade	I_3_^−^/I^−^	14.53	0.727	65.26	6.89	[102]
TiN(S)-PEDOT:PSS	Doctor blade	I_3_^−^/I^−^	14.35	0.724	59.48	6.18	[102]
PEDOT-Ni_3_(PO_4_)_2_	Spin-coating	I_3_^−^/I^−^	12.21	0.746	70.3	6.412	[103]
PEDOT-Co_3_(PO_4_)_2_	Spin-coating	I_3_^−^/I^−^	12.05	0.724	69.6	6.109	[103]
PEDOT-Ag_3_(PO_4_)_2_	Spin-coating	I_3_^−^/I^−^	11.08	0.762	44.1	3.731	[103]
rGO/PPy/PEDOT	APS oxidation/ potentiostatic deposition	I_3_^−^/I^−^	17.0	0.76	55.0	7.1	[104]
PEDOT:PSS/PPy	Electrochemical polymerization	I_3_^−^/I^−^	14.27	0.75	71.0	7.60	[105]

**Table 5 polymers-11-00143-t005:** Photovoltaic performance parameters of DSSCs based on the different polymer electrolyte.

Electrolyte System	*J_SC_* (mA cm^−2^)	*V_OC_* (V)	*FF* (%)	*η* (%)	Ref.
Epichlomer-16, 0.3 g elastomer, 25 mL acetone, 0.03 g NaI	4.20	0.82	47	1.6	[118]
1.4 g polyacrylonitrile, 10 g ethylene carbonate, 5 mL propylene carbonate, 5 mL acetonitrile, 1.5 g NaI, 0.1 g I_2_	3.40	0.58	67.0	4.40	[119]
17.5 wt % Poly(acrylonitrile-*co*-styrene), 0.5 M *N*-methyl pyridine iodide, 0.05 M iodine, ethylene carbonate:propylene carbonate	7.82	0.708	56.0	3.10	[120]
40 wt % PEG, 60 wt % PC, 0.65M KI, 0.065 M I_2_,	14.89	0.73	66.45	7.22	[121]
10 wt % PEO, 0.1 M LiI, 0.1 M I_2_, 0.6 M DMPII, 0.45 M NMBI in MePN	9.13	0.76	68.1	4.72	[122]
2 wt % PS, 0.6 M butylmethylimidazolium iodide, 0.03 M I_2_, 0.1 M guanidinium thiocyanate, 0.5 M *t*BP in acetonitrile/valeronitrile (85:15, *V*/*V*)	15.30	0.77	64.0	7.54	[123]
*N*-methyl pyridine iodide, iodine, γ-BL, Triton X-100, 0.5 mL glacial acetic acid, 2 mL titanium isopropoxide	5.63	0.635	51.4	3.06	[124]
PAA-PEG, 0.5 M NaI, 0.05 M I_2_, 0.4 M pyridine, 30 vol % NMP, 70 vol % GBL	15.28	0.661	62.4	6.30	[126]
PAA-PEG, 1.0 M NaI, 0.15 M I_2_, 0.4 M pyridine, 30 vol % NMP, 70 vol % GBL	11.41	0.724	63.5	5.25	[126]
PEG:LiI/I_2_ + 15 wt % PEGDA	---	---	---	4.18	[127]
PPDD, 0.5 M iodide, 0–0.5 M inorganic salt, 0.5 M NMBI	17.10	0.70	64.0	7.72	[128]
PAA-PEG 20,000, 0.5 M NaI, 0.05M I_2_, 30 vol % NMP, 70 vol % GBL, 0.4 M PY, DMF	12.55	0.735	66.1	6.10	[129]
PEO-TiO_2_(I^−^/I_3_^−^)	7.20	0.664	58.0	4.20	[130]
PEO/TiO_2_/I^−^/I_3_^−^	2.05	0.67	39.0	0.96	[131]
9 wt % SiO_2_ nanorod/PMII/I_2_/PEGDME	12.0	0.621	62.8	4.68	[132]
10 wt % P (VDF-HFP), 0.6 M DMPII, 0.1 M LiI, 0.1 M I_2_, 0.45 M NMBI	14.746	0.621	62.5	5.72	[133]

**Table 6 polymers-11-00143-t006:** Photovoltaic performances parameters of PSCs based on the different polymer HTLs.

Device Structure of PSCs with Polymer HTLs	*J_SC_* (mA·cm^−2^)	*V_OC_* (V)	*FF* (%)	*η* (%)	Ref.
FTO/mp-TiO_2_/CH_3_NH_3_PbI_3_/PTAA/Au	16.40	0.90	61.4	9.0	[189]
FTO/bl-TiO_2_/mp-TiO_2_/FAPbI3-based perovskite/PTAA/Au	24.70	1.06	77.5	20.2	[190]
FTO/TiO_2_/CH_3_NH_3_PbI_3_/S197/Au	17.60	0.967	70.0	12.0	[191]
ITO/PEDOT:PSS/FASnI3/C_60_/BCP/Cu	24.87	0.45	0.63	7.05	[194]
ITO/PEDOT:PSS/CH_3_NH_3_PbI_3_/PC_61_BM/Au/LiF	19.35	0.80	73.0	11.90	[195]
ITO/PEDOT:PSS/CH_3_NH_3_PbI_3_/PCBM/Bphen/Al	22.53	0.983	77.3	17.16	[196]
FTO/(PEDOT:PSS):Nafion/CH_3_NH_3_PbI_3_/PCBM/Al	22.17	1.02	73.47	16.68	[202]
FTO/TiO_2_/CH_3_NH_3_PbI_3_/PANI	14.48	0.78	65.0	7.34	[203]
FTO/TiO_2_/CH_3_NH_3_PbI_3_/PPEDOT	20.08	0.89	69.0	12.33	[204]
FTO/Mp-TiO_2_/CH_3_NH_3_PbI_3_/P3HT/Au	12.60	0.73	73.2	6.70	[189]
FTO/TiO_2_/CH_3_NH_3_PbBr_3_/P3HT/Au	1.13	0.84	54.0	0.52	[206]
ITO/perovskite/P3HT/MoO_3_/Ag	17.90	0.87	66.0	10.80	[207]
ITO/perovskite/P/MoO_3_/Ag	13.80	0.81	52.0	5.62	[207]
FTO/bl-TiO_2_/CH_3_NH_3_PbI_3_/P3HT+Li-TFSI+tBP/Au	20.10	0.92	74.0	13.70	[208]
FTO/Mp-TiO_2_/CH_3_NH_3_PbI_3_/P3HT+BCN/Au	17.75	0.83	49.0	7.60	[209]
FTO/Mp-TiO_2_/CH_3_NH_3_PbI_3_/P3HT-graphdiyne/Au	19.63	0.939	71.5	13.17	[210]
FTO/Mp-TiO_2_/CH_3_NH_3_PbI_3_/P3HT/SWCNT-PMMA/Au	22.71	1.02	66.0	15.30	[211]
FTO/Mp-TiO_2_/CH_3_NH_3_PbI_3_/PCPDTBT/Au	10.30	0.77	66.7	5.30	[189]
FTO/Mp-TiO_2_/CH_3_NH_3_PbI_3_/PCDTBT/Au	10.50	0.92	43.7	4.20	[189]
FTO/bl-TiO_2_/mp-TiO_2_/CH_3_NH_3_PbI_3_/PDPPDBTE/Au	14.40	0.8553	74.9	9.20	[212]
FTO/bl-SnO2/CH_3_NH_3_PbI_3_/RCP/Au	21.9	1.08	75.0	17.30	[213]

**Table 7 polymers-11-00143-t007:** Photovoltaic performances parameters of OPVs based on the different polymer donors.

Device Structure of OPVs with Polymer Donors	*J_SC_* (mA·cm^−2^)	*V_OC_* (V)	*FF* (%)	*η* (%)	Ref.
ITO/PEDOT:PSS/P3HT:ICBA/Ca/Al	10.61	0.84	72.7	6.48	[240]
ITO/PEDOT:PSS/P3HT:ICBA/PFCn6:K^+^/Ca/Al	11.65	0.89	72.6	7.50	[241]
ITO/PEDOT:PSS/DiO/Al	3.55	1.01	58.0	2.10	[242]
ITO/PEDOT:PSS/APFO-15:PCBM/LiF/Al	6.00	1.00	63.0	3.70	[247]
ITO/PEDOT:PSS/BisEH-PFDTBT:PC_71_BM/Ca/Al	8.40	0.95	44.0	3.50	[248]
ITO/PEDOT:PSS/BisDMO-PFDTBT:PC_71_BM/Ca/Al	9.10	0.97	51.0	4.50	[248]
ITO/PEDOT:PSS/PTB7:PC_71_BM/Ca/Al	14.50	0.74	68.97	7.40	[251]
ITO/PEDOT:PSS/PBDTTT-E:PC_70_BM/Ca/Al	13.20	0.62	63.0	5.15	[252]
ITO/PEDOT:PSS/PBDTTT-C:PC_70_BM/Ca/Al	14.70	0.70	64.1	6.58	[252]
ITO/PEDOT:PSS/PBDTTT-CF:PC_70_BM/Ca/Al	15.20	0.76	66.9	7.73	[252]
ITO/PEDOT:PSS/PTB7:PC_71_BM/PFN interlayer/Al	15.40	0.759	70.6	8.24	[253]
ITO/PFN interlayer/PTB7:PC_71_BM/MoO_3_/Al	17.20	0.740	72.0	9.15	[253]
ITO/PEDOT:PSS/PIDT-DFBT:PC_71_BM/Ca/Al	11.20	0.97	55.0	5.97	[254]
ITO/PEDOT:PSS/PIDTT-DFBT:PC_71_BM/Ca/Al	12.21	0.95	61.0	7.03	[254]
ITO/PEDOT:PSS/PBDTBDD:PC_61_BM/Ca/Al	10.68	0.86	72.27	6.67	[255]
ITO/PEDOT:PSS/PDBT-T1:PC_70_BM/ZrAcac/Al	14.11	0.92	75.0	9.74	[256]
ITO/PEDOT:PSS/PBDTTPD:PC_71_BM/LiF/Al	9.10	0.87	53.8	4.20	[260]
ITO/PEDOT:PSS/PBDTBTz:PC_70_BM/Al	7.84	0.86	57.0	3.82	[261]
ITO/ZnO/PBDB-T:ITIC/MoO_3_/Al	16.81	0.899	74.2	11.21	[263]
ITO/ZnO/PFN-Br/PBDB-T:F-M/PEDOT:PSS/ZnO/ PTB7-Th:O6T-4F:PC_71_BM/MoO_3_/Ag	14.35	1.642	73.7	17.36	[264]
ITO/PEDOT:PSS/PBDTT-DPP:PC_71_BM/Ca/Al	14.00	0.73	65.0	6.60	[265]
ITO/PEDOT:PSS/PBDTP-DPP:PC_71_BM/Ca/Al	13.60	0.76	60.0	6.20	[265]
ITO/PEDOT:PSS/PBDTT-FDPP:PC_71_BM/Ca/Al	13.80	0.77	55.0	5.80	[265]
ITO/PEDOT:PSS/PBDTP-FDPP:PC_71_BM/Ca/Al	7.42	0.77	59.0	3.30	[265]
ITO/PEDOT:PSS/PDTTDPP:PC_71_BM/LiF/Al	13.70	0.66	66.1	6.05	[265]

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
