# Peer review of "The Applications of Polymers in Solar Cells: A Review"

_polymers, 2019, doi:10.3390/polym11010143_

Round 1

Reviewer 1 Report

The Manuscript “The application of polymers in solar cells: a review” faces an important and vast issue to try to gather in an organized way the existing information on this topic.

The work is carried out on a variety of systems and is digging into important, meaningful and up to date details.

Nevertheless, the work lacks homogeneity of appropriate depth and precision in some of the fields.

The introduction is also lacking grip in correspondence of the same fields.

A revision of the manuscript is necessary by the authors to eliminate the gaps that in some case are so severe to make to doubt of the scientific soundness of the script.

I list here some examples and some advices: the observations should be extended to the rest of the manuscript

Line 14 : abundant.  Here it has no meaning. The same word is abused in many other parts of the manuscript.

36-37 environmental pollution process : meaningless.

40 and cadmium is toxic.

43 simple ? Seriously?

54-58 Nonsense, what do you mean by monomer?

64-65 Nonsense.

71 this chapter is not well organized: there is a detailed description of DSSCs but it is not finalized to the description of the role of polymers so it should be equilibrated.

87 what is components materials?

88 innate?

120 The polymers are not sintered but something else happens to them

126 probably too many in the approximated number.

316-318 the meaning of the sentence is lost.

345 The cation effect on the viscosity of the polymer electrolyte is perhaps a topic that could referred to (adding appropriate references).

351-352 unclear.

374 PEG is a common name of that polymer so the article the is not necessary.

396-399 other more recent bibliography might be considered.

454 the meaning of the word “hybrid “is here as in many other places ambiguous or wrong in this case, very wrong.

460-470 Poor description.

479 incorporating PEG into.

483 that PEG.

491 large abuse of the article “the”.

772 hybrids in which meaning? ; fabricate platinum.

773 Why extremely? Abundant?

774-776 Unclear

779 Conductive? Really wrong meaning.

790-792 Unclear.

Author Response

Response to Reviewers

Manuscript ID: polymers-406443
Title: The applications of polymers in solar cells: a review

Journal: Polymers

We are very pleased to hear from you. Thank you very much for your comments to our paper (polymers-406443)! Those comments are all valuable and very helpful for revising and improving our manuscript, as well as the important guiding significance to our researches. According to your comments, a detailed revision is made, and the corrected sentences and words are highlighted in red type in the revised manuscript. The point to point explanations are shown as below:

Reviewers' comments:

Reviewer 1

The Manuscript “The application of polymers in solar cells: a review” faces an important and vast issue to try to gather in an organized way the existing information on this topic. The work is carried out on a variety of systems and is digging into important, meaningful and up to date details. Nevertheless, the work lacks homogeneity of appropriate depth and precision in some of the fields. The introduction is also lacking grip in correspondence of the same fields. A revision of the manuscript is necessary by the authors to eliminate the gaps that in some case are so severe to make to doubt of the scientific soundness of the script. I list here some examples and some advices: the observations should be extended to the rest of the manuscript.

Reply: We are highly thankful to the esteemed reviewer for taking your valuable time in reading our manuscript, and providing kind and constructive comments for improvement of the manuscript. According to your comments, we have revised the manuscript carefully.

1. Line 14 : abundant. Here it has no meaning. The same word is abused in many other parts of the manuscript.

Reply: (1) Polymers can be used to adjust the device components and structures of these solar cells purposefully due to their abundant properties.” has been revised as  Polymers can be used to adjust the device components and structures of these solar cells purposefully due to their diversified properties.”

(2) “The solar energy stands out owing to solar is abundant reserves, environmentally friendly and free from regional restrictions” has been revised as “The solar energy stands out owing to solar is environmentally friendly and free from regional restrictions”.

(3) “The abundant functional groups make the polymers produce the interactions with the perovskite grains”has been revised as The diversified functional groups make it possible for the polymers to produce the interactions with the perovskite grains.

(4) And the more abundant pore structure is, the larger specific surface area is has been revised as And the more pore structure is, the larger specific surface area is.

(5) Except PPy, the polyaniline (PANI) is also the most studied conductive polymers due to its relatively low cost, abundant oxidation states with different colors, and acid/base doping responsiveness. has been revised as “Except PPy, the polyaniline (PANI) is also the most studied conductive polymers due to its relatively low cost, multiple oxidation states with different colors, and acid/base doping responsiveness.” 

(6) “The abundant active sites, good contact performance with substrate, and the oriented electron transfer along the nanoribbons equip the PANI NR based DSSC a PCE of 7.23%” has been revised as “The lots of active sites, good contact performance with substrate, and the oriented electron transfer along the nanoribbons equip the PANI NR based DSSC a PCE of 7.23%”.

(7) “The abundant functional groups make PEO and PEG enhance the ionic conductivity by forming the interaction with the alkali metal cations” has been revised as “The various functional groups make PEO and PEG enhance the ionic conductivity by forming the interaction with the alkali metal cations”.

2. 36-37 environmental pollution process : meaningless.

Reply: “environmental pollution process” at line 36-37 has been deleted.

3. 40 and cadmium is toxic.

Reply: “Although these thin-film solar cells have high efficiency and stability, the abundance of gallium and indium in the crust is low and the cadmium is toxic.” has been revised as “Although these thin-film solar cells have high efficiency and stability, the abundance of gallium and indium in the crust is low and cadmium is toxic”.

4. 43 simple ? Seriously?

Reply: “This kind solar cells have the advantages of lightweight, low cost, and simple preparation has been revised as “This kind solar cells have the advantages of lightweight and low cost.

5. 54-58 Nonsense, what do you mean by monomer?

Reply: The description about monomer polymerized into polymers has been deleted. And “Under certain conditions, the monomer can be assembled into the polymers with three-dimensional network structure, which decides the polymers can be employed as the template in mesoporous materials or polymeric matrix in the solid electrolyte” has been revised as The three-dimensional network structures of polymers decides that they can be employed as the template to fabricate mesoporous materials or be used as polymeric matrix in solid electrolyte.

6. 64-65 Nonsense.

Reply: “Because the silicon-based solar cells and thin-film solar cells are mainly based on the inorganic semiconductors, the applications of polymers in solar cells are mainly concentrated on the emerging solar cells, including the DSSC, PSC, and OPV.” has been revised as “In this review, the applications of polymers in solar cells are mainly concentrated on DSSC, PSC, and OPV. 

7. 71 this chapter is not well organized: there is a detailed description of DSSCs but it is not finalized to the description of the role of polymers so it should be equilibrated.

Reply: According to your comments, the corresponding contents have been reorganized to stick to the topic. And the revised contents are as follows:

Polymers in dye-sensitized solar cell (DSSC)

Inspired by the photosynthetic process in plants, the O’Regan and Grätzel proposed the first prototype of DSSC in 1991 [27]. The DSSC has achieved stunning progress in recent years due to its low cost, excellent efficiency, and handy and eco-friendly fabrication process [28]. In DSSC, the electrolyte was encapsulated between the anode and the counter electrode to form a sandwich structure. The principle of DSSC was shown in Figure 2 [29−33]. Generally, the electrons at excited state inject into the conduction band (CB) of semiconductor, and then flow to the counter electrode through the external circuit; The oxidized electrolyte obtains the electrons from the counter electrode to produce the reduced electrolyte, realizing the electrolyte regeneration; The electron recombinations occur at these three parts and their interfaces. In this sense, the three parts of the DSSC play the imperative and indispensable roles in deciding the device performances. As we all known, the choosing and fabricating of components materials are crucial. The polymers can be applied into fabricating the components materials of DSSC due to their various properties and relatively good stability, such as: the polymers can be used to fabricate flexible substrates, to constitute mesoporous structure in anode, to prepare polymer gel electrolyte, and to catalyze electrolyte reduction as counter electrodes. The applications of polymers in DSSC are illustrated in the following sections in detail.

8. 87 what is components materials?

Reply: We apologize for our unclear description. What we want to describe is the constituent materials. As the shown in the answer of question 8, the “components materials” have been revised as constituent materials. And the constituent materials of DSSC refers to the substrates, photoanode, electrolyte and counter electrode .

9. 88 innate?

Reply: We apologize for our inappropriate description. And The polymers can be applied into fabricating the constituent materials of DSSC due to their various properties and innate stability has been revised as “Due to the various properties, the polymers can be applied into fabricating the constituent materials, such as: the polymers can be used to fabricate flexible substrates, to constitute mesoporous structure in anode, to prepare polymer gel electrolyte, and to catalyze electrolyte reduction as counter electrodes”.

10. 120 The polymers are not sintered but something else happens to them.

Reply: We apologize for our unclear description. The polymers were sintered to  form mesoporous structure. The related descriptions in our manuscript have been revised as: Hou et al. fabricated a series of mesoporous TiO2 anodes by annealing the blade-coated TiO2 films, which contains with defferent content  polyvinylpyrrolidone (PVP).

11. 126 probably too many in the approximated number.

Reply: After careful checking and calculation, We confirm that the approximated number is correct. To avoid ambiguity, I have supplemented device efficiency that without PVP: The DSSC device based on TiO2 anode with 20 wt% of PVP achieves the highest conversion efficiency of 8.39%, approximately increased by 56.53% than that of the DSSC fabricated without PVP (5.36%). 

12. 316-318 the meaning of the sentence is lost.

Reply: The mentioned contents “Eventhough the devices based on the liquid electrolytes have obtained the stunning progresses, there are still some practical issues needed to solve.” has been deleted.

13. 345 The cation effect on the viscosity of the polymer electrolyte is perhaps a topic that could referred to (adding appropriate references). 

Reply: “Excessive polymers host will retard the ionic movement by increasing the viscosity of gel electrolyte. Excessive liquid electrolyte will lead to the sealing problem.” The above description in our manuscript has not mentioned the cation effect on viscosity.

14. 351-352 unclear. 

Reply: “(e) Schematic of the conversion from the liquid electrolyte into a gel electrolyte” has been revised as Conversion schematic of liquid electrolyte to gel electrolyte by using valeronitrile solvent to dissolve the PS nanobeads.

15. 374 PEG is a common name of that polymer so the article the is not necessary. 

Reply: “After treating by the UV light illumination, the poly(ethylene glycol) (PEG) and poly(ethyleneglycol) diacrylate (PEGDA) formed the crosslinked gel structure without damaging the device components.” has been revised as “After treating by the UV light illumination, poly(ethylene glycol) (PEG) and poly(ethyleneglycol) diacrylate (PEGDA) formed the crosslinked gel structure without damaging the device components.”. the has been deleted.

16. 396-399 other more recent bibliography might be considered. 

Reply: The recent bibiography about all-weather solar cells have been revised as Re 136-137.

17. 454 the meaning of the word “hybrid “is here as in many other places ambiguous or wrong in this case, very wrong. 

Reply: “The H atom in MA+ or FA+ can form the hydrogen bonds with special hybrid atoms.” In this case, the hybrid atoms refer to the atoms with high electronegativity and small radius, such as F, O, and N atoms. Therefore, we have revised the sentences as:The H atom in MA+ or FA+ can form the hydrogen bonds with atoms, which possesses high electronegativity and small radius, such as F, O, and N atoms. 

18. 460-470 Poor description.    

19. Reply: The related content has been revised to avoid poor description. And the revised description is as follows:

Besides, the good solubility of polymers in polar solvent reduces the contact angle, which guarentees the spread of precursor solution and the even coverage of perovskite film  [167]. These unique features make the polymers can be used as additives to improve the crystallization morphology and device performances. The Figure 9 shows the molecular structures of common polymers additives, involving the poly(ethyleneglycol) (PEG), polyetherimide (PEI), polyvinylpyrrolidone (PVP), polystyrene (PS), poly (methyl methacrylate) (PMMA), 2,4-dimethyl-poly(triarylamine) (PTAA), Poly(2-ethyl-2-oxazoline) (PEOXA), poly[2-methoxy-5-(2-ethylhexyloxy)-1,4-phenylenevinylene] (MEH-PPV), poly-acrylonitrile (PAN), and poly[(9,9-bis(3'-(N,N-dimethylamino)-propyl)-2,7-fluo-

rene)-alt-2,7-(9,9-dioctyl-fluorene)] (PFN), etc [164−175]. The Figure 10 presents the influences of several polymer additives on the surfacial and cross-sectional morphology of perovskite films. It is obvious that theses polymer additives improve the coverage and reduce the pinholes, which are conducive to reduce recombination and enhance device performance.

20. 479 incorporating PEG into. 

Reply: Chang et al. fabricated the CH3NH3PbI3-xClx perovskite with improved coverage by incorporating the PEG additive into the precursor solution has been revised as Chang et al. fabricated the CH3NH3PbI3-xClx perovskite with improved coverage by incorporating PEG additive into the precursor solution.

21. 483 that PEG.  

Reply: “Zhao et al. found that the PEG also served as the scaffold for perovskite film due to its three-dimensional network structure (Figure 11 a-d)” has been revised as “Zhao et al. found that PEG also served as the scaffold for perovskite film due to its three-dimensional network structure (Figure 11 a-d)”.

22. 491 large abuse of the article “the”.

Reply: We have checked the whole manuscript to avoid large abuse of the. And the unnecessary the have been deleted. And corresponding modification traces have been retained.

23. 772 hybrids in which meaning? ; fabricate platinum. 

Reply: We apologize for our abuse of hybrids. And “Besides, the conductive polymers and corresponding hybrids can be used to fabricate the platinum-free counter electrode materials” has been revised as “Besides, the conductive polymers and corresponding composites can be used to fabricate the platinum-free counter electrode materials”.

24. 773 Why extremely? Abundant?  

Reply: The extremely long chain structure and abundant functional groups also decide the polymers can be used to solidify the liquid electrolyte and form the quasi-solid-state electrolytes. In this case, the “extremely” has been deleted, and the abundant have been revised as various. And the revised sentence is: The long chain structure and containing functional groups also decide the polymers can be used to solidify the liquid electrolyte and form the quasi-solid-state electrolytes.

25. 774-776 Unclear  

Reply: We apologize for our unclear description. “The application researches about the polymers in substrates, photoanodes preparation, and counter electrodes are extensive and mature has been revised as “The applications of polymers in flexible PAN or PET substrates, mesoporous TiO2 photoanodes preparation, polymer get electrolyte and platinum-free counter electrodes are extensive and mature .

26. (759)779 Conductive? Really wrong meaning. 

Reply: “Hence, improving the conductivity of quasi-solid-state electrolytes is still challenging, which can be realized by adjusting the ratios of polymers and liquid electrolyte or by adding the conductive substance into the matrix of polymer. has been revised as “Hence, improving the conductivity of quasi-solid-state electrolytes is still challenging, which can be realized by adjusting the ratios of polymers and liquid electrolyte.

27. 790-792 Unclear. 

Reply: “Besides, it is promising to select the polymers with complementary light-harvesting properties to fabricate the tandem solar cells.” has been revised as “Besides, it is promising to select the polymers with complementary spectrum absorption range to fabricate the ternary or tandem OPVs.”

Thanks very much for taking your time to review our manuscript! The manuscript has been revised according to your comments. If there are any problems, do not hesitate to contact with me. We look forward to your positive response.

Sincerely Yours,

Prof. Yaoming Xiao

Institute of Molecular Science

Shanxi University, Taiyuan 030006, P. R. China

Reviewer 2 Report

The present manuscript entitled "The applications of polymers in solar cells: a review'' reports the key role of polymers into various solar cells architectures.

The manuscript is well-written and provides a detailed review article in the field of solar cells. However, I consider some details are missing from the manuscript contents and therefore I recommend it to be published to Polymers, after minor revision.

For the publication, the authors have to address my comments, as described below.

1)   The authors have to add a section regarding the polymers’ use as ETL in the case of perovskite solar cells. It is well-known that there are polymers (e.g. PFN) that are used as electron transport materials in the normal planar structure PeSCs. Moreover, there are some polymers that are used as interlayers, or passivation layers into PeSCs. They should be also mentioned. Thus, the respective parts in the main text should be also corrected (e.g. in the abstract: “In perovskite solar cells……… The polymers can also be used as the hole transfer materials …….”).

2)   In OPVs, polymers are also used as buffer layers and not only as donors as the authors claim.

3)   The authors do not mention anything about the use of polymers in the case of ternary OPVs, as the third component.

4)   The respective permissions for reproduction must be obtained by other copyright holders. The authors must cite the copyright holders at the end of the Figure captions, according to MDPI instructions.

Author Response

Response to Reviewers

Manuscript ID: polymers-406443
Title: The applications of polymers in solar cells: a review

Journal: Polymers

We are very pleased to hear from you. Thank you very much for your comments to our paper (polymers-406443)! Those comments are all valuable and very helpful for revising and improving our manuscript, as well as the important guiding significance to our researches. According to your comments, a detailed revision is made, and the corrected sentences and words are highlighted in red type in the revised manuscript. The point to point explanations are shown as below:

Reviewers' comments:

Reviewer 2

The present manuscript entitled "The applications of polymers in solar cells: a review'' reports the key role of polymers into various solar cells architectures.

The manuscript is well-written and provides a detailed review article in the field of solar cells. However, I consider some details are missing from the manuscript contents and therefore I recommend it to be published to Polymers, after minor revision.

Reply: We are highly thankful to the esteemed reviewer for taking your valuable time in reading our manuscript, and providing kind and constructive comments for improvement of the manuscript. According to your comments, we have revised the manuscript carefully.

For the publication, the authors have to address my comments, as described below.

1-1)The authors have to add a section regarding the polymers’ use as ETL in the case of perovskite solar cells. It is well-known that there are polymers (e.g. PFN) that are used as electron transport materials in the normal planar structure PeSCs.

1-2)Moreover, there are some polymers that are used as interlayers, or passivation layers into PeSCs. They should be also mentioned. Thus, the respective parts in the main text should be also corrected (e.g. in the abstract: “In perovskite solar cells……… The polymers can also be used as the hole transfer materials …….”).

Reply: We apologize for our incomplete content. In previous manuscript, we ignored the utilizations of polymers as the electron transfer layer. The contents about polymers serve as interlayers or passivation layers into PSCs have been included in our original manuscript. But, the corresponding contents have not been discussed separately and detailedly. As the following shown, the corresponding contents have been supplemented. And the related descriptions have been revised in the manuscript.

1-1)Polymer as the ETL of PSC

In perovskite solar cells, the ETL plays the roles of collecting electron from perovskite films and transporting it into the external circuit. Therefore, an ideal ETL material should have high electron mobilities and matched energy level with perovskite. In regular n-i-p architecture, the common ETL materials are TiO2 and ZnO. In inverted p-i-n architecture, fullerene and corresponding derivatives have been used as the most common ETL materials. However, the temporal stability of fullerene and corresponding derivatives is also poor, which can be evident from the morphological changes upon recrystallization [214]. Therefore, the research about ETL is still in a beginning stage, especially for the non-fullerene ETL. In recent years, the organic polymers have been widely used as the electron transfer materials due to their tunable bandgap, excellent film-forming ability, good electron mobility, and low fabrication cost. The existing organic polymer ETLs are mainly based on the naphthalene diimide (NDI) and perylene diimide (PDI) cores. Naphthalene diimide (NDI) was chosen as it is one of the most explored electron deficient building block for n-type polymers. About NDI-based polymers ETL, Sun et al. developed the PFN-2TNDI ETL by introducing amine functional group to the side chains of fluorine unite of PF-2TNDI. The device assembled with PFN-2TNDI ETL exhibited the PCE of 16.7%, higher than that of PCBM based device (12.9%). The increased device performances comes down to two reasons: the amines on polymer side chains can passivate the surface traps of perovskite; the amines can reduce the work function of metal cathode by forming dipoles at interfaces [215]. Wang et al. have employed poly{[N,N’-bis(2-octyldodecyl)-1,4,5,8-naphthalenediimide-2,6-diyl]-alt-5,5’(2,2’-bithiophene)} (N2200) as the ETL of perovskite and obtained decent PCE of 8.15%, which is competitive to that of PCBM ETL (8.51%). Moreover, in order to prove the universality of organic polymers as ETL in perovskite solar cells, other polymers PNVT-8 and PNDI2OD-TT have also been tested and achieve the efficiency of 7.13% and 6.11%, respectively [216]. Kim et al. developed another P(NDI2DT-TTCN) copolymer ETL with NDI and dicyano-terthiophene. With the intramolecular interaction between the S-N groups, the dicyano-terthiophene shows a pseudo-planar structure, which decides the good electron transport ability and deep LUMO level of P(NDI2DT-TTCN). The device based on P(NDI2DT-TTCN) showed the PCE of 17.0%, outperforming than that of PCBM ETL-based device (14.3%). This study also illustrate that P(NDI2DT-TTCN) ETL is hydrophobic and can passivate the surface against ambient conditions. The P(NDI2DT-TTCN) ETL also exihibited better mechanical stability than PCBM-based ETLs in bending cycles, which indicates a better prospect for flexible PSC [217]. Guo et al. designed a series of PDI-based polymer PX-PDIs ETL with different X copolymerized units: vinylene (V), thiophene (T), selenophene (Se), dibenzosilole (DBS), and cyclopentadithiophene (CPDT). The device based on PX-PDI ETL shows the highest PCE of 10.14%, which can be attributed to that PV-PDI ETL shows the deeper LUMO energy level and better planar structure than other PX-PDI (X=T, Se, DBS, and CPDT) [218]. All these researches provide valuable design guidelines for devising new polymeric ETLs.

1-2)Polymer as the interlayer of PSC

The perovskite solar cells present the layer-by-layer structure. The invert p-i-n architecture  includes the FTO/HTL, HTL/perovskite, perovskite grain/grain, perovskite/ETL, ETL/Au, and Au/atmosphere interfaces. Charge extraction and separation occur at the interfaces, which could suffer from the recombination due to any possible interfacial defects. Therefore, interfacial chemical interaction is decisive factor for the device performances. The interface engineering has been used as a common method to regulate the morphologies of perovskite films, reduce the defects, and improve the device performances. The polymer interfacial modification has been regarded as one of the most effective methods of interface engineering. At the HTL/perovskite interface, Malinkiewicz et al. introduce the poly(N,N’-bis(4-butylphenyl)-N,N’-bis(phenyl)benzidine) (polyTPD) layer, which can transfer hole and block electron due to its LUMO is closer to the conductive bandgap energy [219]. Lin et al. introduced the DPP-DTT, PCDTBT, P3HT, and PCPDTBT layers at the PEDOT:PSS/perovskite interface. And the perovskite films grown on PCDTBT shows the highest crystallinity intensity and device efficiency than others due to its matched energy level [220]. Wen et al. used the insulating PS to treat the initial perovskite films and formed the tunnelling junction between the perovskite film and hole transfer film. The PS films is capable of accelerating the separation of photo-generated electrons and holes by selectively transfer hole while blocking electrons, improving the device efficiency from 15.90% to 17.80% [221]. At the interfaces between perovskite grains, Wang et al. used the chlorobenezene solution of PMMA to treat the as-prepared perovskite films, which can passivate the surface trap states in perovskite films and suppress the recombination [222]. It is worth stressing that manipulating the perovskite films with PMMA is beneficial to enhance the device stability by protecting the perovskite films from oxygen and moisture. At the perovskite/ETL interface, Wang et al. choosed the PS, Teflon, and polyvinylidene-trifluoroethylene copolymer (PVDF-TrFE) as the tunneling materials. The  tunneling materials allow electron transfer from perovskite to C60 ETL and blocks the holes. And the device based on the PS interlayer given the highest PCE of 20.3%, higher than that of control devices (16.9%) [223]. At the ETL/metal interface, polyethyleneimine (PEIE) and poly[3-(6-trimethylammoniumhexyl) thiophene] (P3TMAHT) create surface dipole. The durface dipole make the work function of Ag decrease from original 4.70 eV to 3.97eV (PEIE) and 4.13 eV (P3TMAHT), and the corresponding PCE increase from original 8.53% to 12.01% (PEIE) and 11.28 (P3TMAHT), respectively [224]. These studies provide a guideline for using the polymers to realize the interface regulation and modification.    

2)  In OPVs, polymers are also used as buffer layers and not only as donors as the authors claim.

Reply: The contents about polymers as buffer layers have been supplemented.

Polymers as buffer layer of OPV

In organic photovoltaivs, there is a barrier between the active layer and the electrode. The barrier hinders the quick charge transfer, leading to serious charge accumulation at the interfaces. The charge accumulation increase the recombination probability, and attenuate the device performances. Therefore, it is imperative to accelerate the charge separate and transfer efficiency. In order to solve this problem, an interface buffer layer is introduced in the OPVs, including the anode buffer layer and cathode buffer layer. These buffer layer is often established between the active layer and the electrode to form an ohmic contact, which is conducive to the efficient charge extraction and separation. Besides, the buffer layer also can inhibit the possible diffusion and reaction between the active layer and the electrode material. The polymer serves as one kind of the most effective buffer layers. As the Table 7 summarized, the PEDOT:PSS has been widely used as the most common anode buffer layer to extract holes, and the corresponding devices achieve enhanced efficiency. Despite of the effective charge extraction ability, the acidic PEDOT:PSS solution will cause a certain degree of corrosion of ITO substrate, which in turn reduces the device stability [270]. For the cathode buffer layers, He et al. have applied the alcohol/water-soluble poly [(9,9-bis(3’-(N,Ndimethylamino)propyl)-2,7-fluorene)-alt-

2,7-(9,9-dioctylfluorene)] (PFN) to extract electron in organic photovoltaics [271]. The OPVs assembled with the PFN cathode buffer layers show enhanced JSC, VOC, and FF, which results in a PCE of 8.37% and 6.79% for PTB7 and PCDTBT devices. The possitive effect of PFN buffer layer on device performances include: forming an enhanced built-in potential across the device; and improving the charge separation and transfer efficiency. Na et al. use polyfluorene derivative (WPF-oxy-F) as the interfacial layer between the active layer and metal cathode, which will reduce the metal work function for better electron transport [272]. The PF-EP was introduced as a cathode buffer layer to improve the charge collection efficiency by prevent the metal atoms migrating into the active layer [273].

3)  The authors do not mention anything about the use of polymers in the case of ternary OPVs, as the third component.

Reply: The contents about the applications of polymers in ternary OPVs have been supplemented.

Polymer in ternary OPVs

The traditional organic solar cells are composed of polymer donors and fullerene-based acceptors. Despite of high electron mobilities, fullerene and its derivatives always show large band gap and narrow absorption window, which hinder the further development of OPVs. Therefore, a series of nonfullerene acceptor (NFA) materials have been reported due to its strong light absorption and easily tunable energy levels. Wang et al. introduce a non-fullerene small molecule m-ITIC into PTB7-Th:TQ1 system to fabricate the ternary OPV [267]. The m-ITIC can broaden the spectrum absorption range of OPV, enhancing the resultant device performances further. However, the electron mobilities of NFAs are always low, which affect the device performance directly. Hence, the fullerene acceptor was introduced into the binary polymers donor: nonfullerene acceptor system to fabricate the ternary OPV. As the third component, the fullerene can compensate the deficiencies of nonfullerene acceptors. Fu et al. fabricated the ternary OPVs using wide-bandgap PTB1-C as donor and highly conductive PC71BM and narrow-bandgap nonfullerene IT-2F as acceptor [268]. The resultant device obtained increased PCE, which benefits from the increased light harvesting from of IT-2F and enhanced electron transport. Meanwhile, the addition of PC71BM can also suppress the aggregation of IT-2F, reducing the trap density and trap-assisted recombination. When it comes to the ternary OPVs, it is necessary to consider possible and unfavorable large-scale phase separation. Therefore, it is difficult but imperative to control the morphology while pursuing high performance. Xie et al. reported that ITIC-2Cl molecules could easily form granular aggregates in the binary blend [269]. After adding 20% ICBA into the blend, the aggregation of ITIC-2Cl reduced, which can be attributed to the amorphous ICBA can disturb intermolecular π-π interactions of ITIC-2Cl. The optimized morphology, low LUMO level, and complementary absorption decide a enhanced PCE of 13.4%.    

4) The respective permissions for reproduction must be obtained by other copyright holders. The authors must cite the copyright holders at the end of the Figure captions, according to MDPI instructions.

Reply: The respective permissions for reproduction have been obtained by the corresponding copyright holders. And the copyright holders have been cited at the end of the Figure captions.

Thanks very much for taking your time to review our manuscript! The manuscript has been revised according to your comments. If there are any problems, do not hesitate to contact with me. We look forward to your positive response.

Sincerely Yours,

Prof. Yaoming Xiao

Institute of Molecular Science

Shanxi University, Taiyuan 030006, P. R. China

Reviewer 3 Report

The manuscript by W. Hou et al. presents a comprehensive review of applications of various polymers in solar cells. The authors specifically discuss the utility of various polymers for dye sensitized solar cells, perovskite solar cells and organic solar cells. The review is quite thorough and covers various aspects of polymers for enhancing the performance of solar cells. The manuscript can be recommended for publication in Polymers journal after the following concern is addressed by the authors.

Polymers are not only used as additives for active layers or electron/hole transport layers. The role of polymer-based micro/nanostructures in improving the light trapping capability of solar cells cannot be ignored and should be discussed by the authors. For example, polymer based microlenses and patterned nanogratings have been shown to immensely improve the power conversion efficiencies of organic and perovskite solar cells. Since the focus of review article is to highlight the role of polymers in improving solar cells, I would recommend the authors to include a separate discussion on this topic in their manuscript. I suggest the authors to refer and include these key references on the role of polymer based structures in enhancing the performance of organic or perovskite solar cells.

1.       Trapping light with micro lenses in thin film organic photovoltaic cells, Optics Express 16, 21608-21615, 2008.

2.       Formation of Nanopatterned Polymer Blends in Photovoltaic Devices, Nano Letters 10, 1302–1307, 2010.

3.       Folded reflective tandem polymer solar cell doubles efficiency, Appl. Phys. Lett. 91, 123514, 2007.

4.       Light management in perovskite solar cells and organic LEDs with microlens arrays, Optics Express 25, 10704-10709, 2017.

5.       Nanostructured interfaces in polymer solar cells, Appl. Phys. Lett. 96, 263109, 2010.

6.       Microlens array induced light absorption enhancement in polymer solar cells,
Phys. Chem. Chem. Phys. 15, 4297-4302, 2013.

7.       Nanophotonic Organic Solar Cell Architecture for Advanced Light Trapping using Dual Photonic Crystals, ACS Photonics 1, 840-847, 2014.  

Author Response

Response to Reviewers

Manuscript ID: polymers-406443
Title: The applications of polymers in solar cells: a review

Journal: Polymers

We are very pleased to hear from you. Thank you very much for your comments to our paper (polymers-406443)! Those comments are all valuable and very helpful for revising and improving our manuscript, as well as the important guiding significance to our researches. According to your comments, a detailed revision is made, and the corrected sentences and words are highlighted in red type in the revised manuscript. The point to point explanations are shown as below:

Reviewers' comments:

Reviewer 3

The manuscript by W. Hou et al. presents a comprehensive review of applications of various polymers in solar cells. The authors specifically discuss the utility of various polymers for dye sensitized solar cells, perovskite solar cells and organic solar cells. The review is quite thorough and covers various aspects of polymers for enhancing the performance of solar cells. The manuscript can be recommended for publication in Polymers journal after the following concern is addressed by the authors.

Reply: We are highly thankful to the esteemed reviewer for taking your valuable time in reading our manuscript, and providing kind and constructive comments for improvement of the manuscript.

Polymers are not only used as additives for active layers or electron/hole transport layers. The role of polymer-based micro/nanostructures in improving the light trapping capability of solar cells cannot be ignored and should be discussed by the authors. For example, polymer based microlenses and patterned nanogratings have been shown to immensely improve the power conversion efficiencies of organic and perovskite solar cells. Since the focus of review article is to highlight the role of polymers in improving solar cells, I would recommend the authors to include a separate discussion on this topic in their manuscript. I suggest the authors to refer and include these key references on the role of polymer based structures in enhancing the performance of organic or perovskite solar cells.

1. Trapping light with micro lenses in thin film organic photovoltaic cells, Optics Express 16, 21608-21615, 2008.

2. Formation of Nanopatterned Polymer Blends in Photovoltaic Devices, Nano Letters 10, 1302–1307, 2010.

3. Folded reflective tandem polymer solar cell doubles efficiency, Appl. Phys. Lett. 91, 123514, 2007.

4. Light management in perovskite solar cells and organic LEDs with microlens arrays, Optics Express 25, 10704-10709, 2017.

5. Nanostructured interfaces in polymer solar cells, Appl. Phys. Lett. 96, 263109, 2010.

6. Microlens array induced light absorption enhancement in polymer solar cells, 
Phys. Chem. Chem. Phys. 15, 4297-4302, 2013.

7. Nanophotonic Organic Solar Cell Architecture for Advanced Light Trapping using Dual Photonic Crystals, ACS Photonics 1, 840-847, 2014.

Reply: Our review article is aiming at highlighting the role of polymers in improving the device efficiency of solar cells. The polymer-based micro/nanostructures (such as microlenses and patterned nanogratings) have been reported to immensely improve the power conversion efficiencies of solar cells. Therefore, we agree to include a separate discussion on this topic in our manuscript. And the corresponding contents have been supplemented in our manuscript.

Polymer-based micro/nanostructures in the OPVs

The power conversion efficiencies (PCE) of organic photovoltaics (OPV) is still limited and can not be compared with their inorganic counterpart. The low light trapping efficiency and low charge carrier mobility are the main limitations.

Enhancing light trapping in OPV is important owing to thin active layer always leads to optical losses accounting for 40% of total losses [274]. In planar heterojunction structure, the common exciton diffusion length is around 5-10 nm, which decides that the thickness of active layer is about 10 nm [275]. In bulk heterojunction structure, the thickness of active layer (about 100~200 nm) is still not enough to absorb and trap light efficiently due to low charge carrier mobilities in most organic materials. Therefore, improving the light trapping within existing OPV architectures is imperative. Apart from developing new materials, another ways of improving the light trapping include the uses of textured substrates, noble metal nanoparticles, tandem solar cells, and microlens arrays (MLA). In the case of textured substrates, the aspect-ratio of textures need to be designed optimally to ensure the conformal coating of active layers [276]. The unreasonable design will result in shunts and recombination [276]. The noble metal nanoparticles doped active layers employ plasmonic near-field enhancement effects to trap more light. However, the excessive metallic and surface stabilizer on metal could serve as the recombination centers to retard charge transport. In terms of tandem polymer solar cells, the folded structure cause light trapping at high angles and large photocurrent density. And the tandem polymer solar cells also allow multiple bandgap solar cells series or parallel connection. Tvingstedt et al. reported a tandem cell by folding two planar but different cells toward each other, where single cells reflect the nonabsorbed light upon another adjacent cell to realize the spectral broadening and light trapping [277]. And the ultimate device performances were enhanced. Designing MLA structures on the backside of the transparent conductive substrates has been demonstrated as a universal method to increase light trapping of OPV. This approach couldn’t exert influences on the device fabrication processing and active layer morphology. Chen et al. has designed the near-hemispherical MLA of 2 mm diameter, which shows the higher ability to refract incoming light than a pure-hemispherical shape [278]. The diameter of this MLA is close to visible light wavelength. This property decides that the near-hemispherical MLA can not only reduce surface reflections, but also can utilize optical interference to enhance light intensity inside the active layer. Ultimately, for the P3HT:PCBM active layer system, the device with near-hemispherical MLA obtained the increased absorption, absolute external quantum efficiency (EQE), and PCE by 4.3%. For the PCDTBT:PC70BM active layer system, near-hemispherical MLA increases the absorption, EQE and PCE by 10.0%. All of these enhancements can attributed to the increased light path and absorption as well as diffraction induced enhanced light intensity. Tvingstedt et al. also studied and demonstrated the principle of MLAs trap the light in detail [279]. The MLAs can make the light display strong directional asymmetric transmission, realizing the recycle of reflected photons. Peer et al. design the device based on dual photonic cryatsls, where polymer MLAs on the glass was coupled with a photonic-plasmonic crystal at the metal cathode on the back of the cell. The MLAs focuses light on the periodic nanostructure, realizing strong light diffraction. The surface plasmon and waveguiding effect of photonic-plasmonic crystal enhance long wavelength absorption. The joint efforts of MLAs and photonic-plasmonic crystal result in absorption enhancement of 49% and photocurrent enhancement of 58% [280]. Except for the OPV, the MLAs have also been extend to improve the device performances of perovskite and organic light emitting diodes [281] .   

Improving the charge carrier mobility and reducing the recombination are imperative to enhance the device efficiency of OPV. For common device with BJH structure, the electron donor and acceptor interspersed randomly in the active layer, which always leads to the discontinuous electron transport pathways. Hence, the internal interfacial area must be large enough to create the continuous pathways for charge separation and transport. In order to solve this problem, He et al. constitute the nanostructured polymer heterojunctions of composition and morphology in active layer through a double nanoimprinting process [282]. The devices based on double-imprinted poly((9,9-dioctylfluorene)-2,7-diyl-alt-[4,7-bis(3-hexylthien-5-yl)-

2,1,3-benzothiadiazole]-2′,2′′-diyl) (F8TBT)/poly(3-hexylthiophene) (P3HT) films obtained a PCE of 1.9%, higher than that of planar F8TBT/P3HT (0.36%). The enhanced device performance can be attributed to ordered structure in active layer can enhance the separation of electron-hole pairs and improve carrier mobilities further. Wiedemann et al. fabricate similar PCBM/P3HT bilayer devices with controlled nanostructured interfaces by combining nanoimprinting and lamination techniques, which presented a higher PCE (0.05%) than that of common bilayer structured device (0.03%) [283]. This method is also suitable for any other polymers combination.

Thanks very much for taking your time to review our manuscript! The manuscript has been revised according to your comments. If there are any problems, do not hesitate to contact with me. We look forward to your positive response.

Sincerely Yours,

Prof. Yaoming Xiao

Institute of Molecular Science

Shanxi University, Taiyuan 030006, P. R. China

Round 2

Reviewer 1 Report

The manuscript has been largely revised, some small correction can be done autonomously by the Authors:

125 "inject" is a transitive verb: make changes accordingly in the sentence.

191 which contain different

435 there is only one type of functional groups: remove "various"

444 here the article is necessary: of the gel electrolyte

584-619 three articles can be removed : The Figure 9, The figure 10 and  the poly(ethyleneglycol)

621 "which" is unfitting here: perhops "and are therefore" is perhaps better

629 removing only the article make teh sentence worse, please ripristinate "the PEG additive" or remove also "additive"

Author Response

Response to Reviewers

Manuscript ID: polymers-406443
Title: The applications of polymers in solar cells: a review

Journal: Polymers

We are very pleased to hear from you. Thank you very much for your comments to our paper (polymers-406443)! Those comments are all valuable and very helpful for revising and improving our manuscript, as well as the important guiding significance to our researches. According to your comments, a detailed revision is made, and the corrected sentences and words are highlighted in red type in the revised manuscript. The point to point explanations are shown as below:

Reviewers' comments:

Reviewer 1

Comments and Suggestions for Authors

The manuscript has been largely revised, some small correction can be done autonomously by the Authors:

Reply: We are highly thankful to the esteemed reviewer for taking your valuable time in reading our manuscript, and providing kind and constructive comments for improvement of the manuscript. According to your comments, we have revised the manuscript carefully.

(1) 125 "inject" is a transitive verb: make changes accordingly in the sentence.

Generally, the electrons at excited state inject into the conduction band (CB) of semiconductor, and then flow to the counter electrode through the external circuit. has been revised as Generally, the electrons at excited state inject the conduction band (CB) of semiconductor, and then flow to the counter electrode through the external circuit. 

(2) 191 which contain different

Reply: Hou et al. fabricated a series of mesoporous TiO2 anodes by annealing the blade-coated TiO2 films, which contains with different content polyvinylpyrrolidone (PVP). has been revised as “Hou et al. fabricated a series of mesoporous TiO2 anodes by annealing the blade-coated TiO2 films, which contain different content  polyvinylpyrrolidone (PVP) ”.

(3) 435 there is only one type of functional groups: remove "various" 

Reply: “The various functional groups make PEO and PEG enhaning the ionic conductivity by forming the interaction with the alkali metal cations and leaving the free-moving iodide anions, thus reducing the recombination and enhancing the device performances” has been revised as “The functional groups make PEO and PEG enhaning the ionic conductivity by forming the interaction with the alkali metal cations and leaving the free-moving iodide anions, thus reducing the recombination and enhancing the device performances”.

(4) 444 here the article is necessary: of the gel electrolyte

Reply: The related baseless description in manuscript has been revised. “Excessive polymers host will retard the ionic movement by increasing the viscosity of gel electrolyte. has been revised as “Excessive polymers host will retard the ionic movement [29,124].

(5) 584-619 three articles can be removed : The Figure 9, The figure 10 and the poly(ethyleneglycol)

Reply: Three articles the in “The Figure 9, The figure 10 and the poly(ethyleneglycol)” has been removed.

(6) 621 "which" is unfitting here: perhops "and are therefore" is perhaps better

Reply: It is obvious that theses polymer additives improve the coverage and reduce the pinholes, which are conducive to reduce recombination and enhance device performance. has been revised as It is obvious that theses polymer additives improve the coverage and reduce the pinholes, and are therefore conducive to reduce recombination and enhance device performance.

(7) 629 removing only the article make the sentence worse, please ripristinate "the PEG additive" or remove also "additive" 

Reply: “Chang et al. fabricated CH3NH3PbI3-xClx perovskite with improved coverage by incorporating PEG additive into the precursor solution  has been revised as “Chang et al. fabricated CH3NH3PbI3-xClx perovskite with improved coverage by incorporating the PEG additive into the precursor solution ”.

Thanks very much for taking your time to review our manuscript! The manuscript has been revised according to your comments. If there are any problems, do not hesitate to contact with me. We look forward to your positive response.

Sincerely Yours,

Prof. Yaoming Xiao

Institute of Molecular Science

Shanxi University, Taiyuan 030006, P. R. China

Reviewer 3 Report

The authors have satisfactorily addressed the concerns of the reviewer and made corresponding revisions in their manuscript. The manuscript can now be accepted for publication in Polymers journal. 

Author Response

Response to Reviewers

Manuscript ID: polymers-406443
Title: The applications of polymers in solar cells: a review

Journal: Polymers

We are very pleased to hear from you. Thank you very much for your comments to our paper (polymers-406443)! Those comments are all valuable and very helpful for revising and improving our manuscript, as well as the important guiding significance to our researches. According to your comments, a detailed revision is made, and the corrected sentences and words are highlighted in red type in the revised manuscript. The point to point explanations are shown as below:

Reviewers' comments:

Reviewer 3

Comments and Suggestions for Authors

The authors have satisfactorily addressed the concerns of the reviewer and made corresponding revisions in their manuscript. The manuscript can now be accepted for publication in Polymers journal. 

Reply: We are highly thankful to the esteemed reviewer for taking your valuable time in reading our manuscript, and providing kind and constructive comments for improvement of the manuscript. 

Thanks very much for taking your time to review our manuscript! The manuscript has been revised according to your comments. If there are any problems, do not hesitate to contact with me. We look forward to your positive response.

Sincerely Yours,

Prof. Yaoming Xiao

Institute of Molecular Science

Shanxi University, Taiyuan 030006, P. R. China
